# Variational Assimilation of Land Surface Temperature within the ORCHIDEE Land Surface Model Version 1.2.6

H. S. Benavides Pinjosovsky[1,2,3], S. Thiria[1], C. Ottlé[2], J. Brajard[1], F. Bradran[1] and P. Maugis[2]

[1]Laboratoire d'Océanographie et du Climat: Expérimentations et Approches Numériques, IPSL Paris, France}

[2]Laboratoire des Sciences du Climat et de l'Environnement, IPSL, CNRS-CEA-UVSQ, Gif-sur-Yvette, France}

[3]CLIMMOD Engineering}, Orsay, France

Correspondence to: H. S. Benavides Pinjosovsky (spinjosovsky@gmail.com) and S. Thiria (sylvie.thiria@locean-ipsl.upmc.fr)

**Abstract.** The SECHIBA module of the ORCHIDEE land surface model describes the exchanges of water and energy between the surface and the atmosphere. In the present paper, the adjoint semi-generator software called YAO was used as a framework to implement a 4D-VAR assimilation scheme of observations in SECHIBA. The objective was to deliver the adjoint model of SECHIBA (SECHIBA-YAO) obtained with YAO to provide an opportunity for scientists and end users to perform their own assimilation. SECHIBA-YAO allows the control of the eleven most influent internal parameters of the soil water content, by observing the land surface temperature or remote sensing data such as the brightness temperature. The paper presents the fundamental principles of the 4D-Var assimilation, the semi-generator software YAO and a large number of experiments showing the accuracy of the adjoint code in different conditions (sites, PFTs, seasons). In addition, a distributed version is available in the case for which only the land surface temperature is observed.

Keywords: Sensitivity Analysis, Data Assimilation, Adjoint model, Land Surface Temperature

## 1. Introduction

Land surface models (LSM) simulate the interactions between the atmosphere and the land surface, which directly influence the exchange of water, energy and carbon with the atmosphere. They are important tools for understanding the main interaction and feedback processes simulating the present climate and making predictions of future climate evolution (Harrison et al., 2009). Such predictions are subject to considerable uncertainties, which are related to the difficulty to model the highly complex physics with a limited set of equations that does not account for all the interacting processes

(Pipunic et al., 2008, Ghent et al. 2011). Understanding these uncertainties is important in order to

obtain more realistic simulations.

A key challenge of a dynamical model is to adjust the output of the model considering an appropriate

source of information. One source of information can be given by measurements (or more generally

observations) that contribute to the understanding of the system evolution (Lahoz et al. 2010). Data

assimilation merges these observations with the dynamical model in order to obtain a more accurate

estimate of the current and future states of the system, given the uncertainties of the model and of the

observations. Two basic methodologies can be used for that purpose. The sequential approach

(Evensen 2003), based on the statistical estimation theory of the Kalman filter, and the variational

approach, the so-called 4DVAR (Le Dimet et al., 1986), built from the optimal control theory

(Robert et al, 2007). It can be proved that both approaches provide the same solution at the end of the

assimilation period, for Gaussian errors (not correlated in time), and linear models. This property

does not stand if the processes under study are nonlinear. The main advantage of 4DVAR comes

from its integration in time achieved during the assimilation of the observations, giving rise to a

global trajectory of the model optimized over the assimilation time window.

Variational data assimilation has been widely used in land surface applications. The assimilation of

land surface temperature (LST) is suitable for an extensive range of environmental problems. As

mentioned in Ridler et al. (2012), LST is an excellent candidate for model optimization since it is

solution of the coupled energy and water budgets, and permits to constrain parameters related to

evapotranspiration and indirectly to soil water content.

In Castelli et al. (1999), a variational data assimilation approach was used to include surface energy

balance in the estimation procedure as a physical constraint (based on adjoint techniques). The

authors worked with satellite data, and directly assimilated soil skin temperatures. They concluded

that constraining the model with such observations improves model flux estimates, with respect to

available measurements. In Huang et al. (2003) the authors developed a one-dimensional land data

assimilation scheme based on an ensemble Kalman filter, used to improve the estimation of the land

surface temperature profile. They demonstrated that the assimilation of LST into land surface models

is a practical and effective way to improve the estimation of land surface state variables and fluxes.

Reichle et al. (2010) performed the assimilation of satellite-derived skin temperature observations

using an ensemble-based, offline land data assimilation system. Results suggest that the retrieved

fluxes provide modest but statistically significant improvements. However, these authors noted

strong biases between LST estimates from *in situ* observations, land modeling, and satellite retrievals

that vary with season and time of the day. They highlighted the importance of taking these biases

into account, otherwise large errors in surface flux estimates can result.

Ghent et al. (2011) investigated the impacts of data assimilation on terrestrial feedbacks of the climate system. Assimilation of LST helped to constrain simulations of soil moisture and surface heat fluxes. Ridler et al. (2012), tested the effectiveness of using satellite estimates of radiometric surface temperatures and surface soil moisture to calibrate a Soil–Vegetation–Atmosphere Transfer (SVAT) model, based on error minimization of temperature and soil moisture model outputs. Flux simulations were improved when the model is calibrated against *in situ* surface temperature and surface soil moisture versus satellite estimates of the same fluxes.

In Bateni et al. (2013), the full heat diffusion equation was employed to perform a variational data assimilation. Deviations terms of the evaporation fraction and a scale coefficient were added as penalization terms in the cost function. Weak constraint was applied to data assimilation with model uncertainty, accounting in this way for model errors. The cost function associated with this experiment contains a term that penalizes the deviation from prior values. When assimilating LST into the model, the authors proved that the heat diffusion coefficients are strongly sensitive. As a conclusion, it can be seen that the assimilation of LST can improve the model simulated flows.

In the present study, we focused on the SECHIBA module (Ducoudré et al. 1993), which is part of the ORCHIDEE Land Surface Model, dedicated to the resolution of the surface energy and water budgets. Our objective was to test the ability of 4DVAR to estimate a set of its inner parameters. A dedicated software (called SECHIBA-YAO) was developed by using the adjoint semi-generator software called YAO developed at LOCEAN-IPSL (Nardi et al. 2009). YAO serves as a framework to design and implement dynamical models, helping to generate the adjoint of the model, which permits to compute the model gradients. SECHIBA-YAO provides an opportunity to control the most influent internal parameters of SECHIBA by assimilating LST (land surface temperature) observations. At a given location and for specific soil and climate conditions, twin experiments of assimilation have been executed. These twin experiments conducted on actual sites were used to demonstrate the accuracy and usefulness of the code and the potential of 4D-VAR when dealing with LST assimilation.

This paper is structured as follows. In Section 2, model and data used to illustrate the capabilities of the SECHIBA-YAO are detailed. In Section 3, fundamentals of variational data assimilation are presented. In addition, principles of YAO and of its associated modular graph formalism are exposed. The principle of the computation of the adjoint with YAO is provided. The implementation of SECHIBA-YAO and the details of the experiments that prove the efficiency of the 4D-Var assimilation, are also given in Section 3. Sensitivity experiments and simple twin experiments at two FLUXNET locations are presented in Section 4. These experiments illustrate the convenience of

YAO to optimize control parameters. Section 5 consists in a discussion and a conclusion. Finally, the specificities of the distributed software are given in Section 6.

## 2. Models and Data

ORCHIDEE is a Land Surface Model developed at the "Institut Pierre Simon Laplace (IPSL)" in France. ORCHIDEE is a mechanistic dynamic global vegetation model (Krinner et al., 2005) representing the continental biosphere and its different biophysical processes. It is part of the IPSL earth system model (Dufresne et al., 2013, and is composed of 3 modules: SECHIBA, STOMATE and LPJ. The version used in this work corresponds to the version 1.2.6, released the 22nd April 2010. SECHIBA computes the water and energy budgets at the biosphere-atmosphere interface, as well as the Gross Primary Production (GPP); STOMATE (Friedlingstein et al., 1999) is a biogeochemical model which represents the processes related to the carbon cycle, such as carbon dynamics, the allocation of photosynthesis respiration and growth maintenance, heterotrophic respiration and phenology and finally, LPJ (Sitch et al., 2003) models the global dynamics of the vegetation, interspecific competition for sunlight as well as fire occurrence. ORCHIDEE has different time scales: 30-minutes for energy and matter, 1-day for carbon processes and 1-year for species competition processes. The full description of ORCHIDEE can be found in Ducoudré et al., 1993, Krinner et al., 2005, d'Orgeval et al., 2006, Kuppel et al., 2012. In the present study, ORCHIDEE 1.9 version is used in a grid-point mode (at a given location), forced by the corresponding local half-hourly gap-filled meteorological measurements obtained at the flux towers. In this study, only the SECHIBA module is considered.

In SECHIBA, the land surface is represented as a whole system composed of various fractions of vegetation types called PFT (Plant Functional Type). A single energy budget is performed at each grid point, but the water budget is calculated for each PFT fraction. The resulting energy and water fluxes between atmosphere, ground and the retrieved temperature represent the canopy ensemble and the soil surface. The main fluxes modeled are the net radiation ($R_n$), soil heat flux ($Q$), sensible ($H$) and latent heat ($LE$) fluxes between the atmosphere and the biosphere, land surface temperature ($LST$) and the soil water reservoir contents. Energy balance is solved once, with a subdivision only for LE in bare soil evaporation, interception and transpiration for each type of vegetation. Water balance is computed for each fraction of vegetation (Plant Functional Type or PFT) present in the grid. The SECHIBA version used in this work models the hydrological budget based on a two-layer soil profile (Choisnel, 1977). The two soil layers represent respectively the surface and the total rooting zone. The soil is considered homogeneous with no sub-grid variability and of a total depth of

$h_{tot} = 2m$. The soil bottom layer acts like a bucket that is filled with water from the top layer. The soil is filled from top to bottom with precipitation; when evapotranspiration is higher than precipitation, water is removed from the upper reservoir. Runoff arises when the soil is saturated. SECHIBA inputs are: $R_{lw}$ the incoming infrared radiation; $R_{sw}$ the incoming solar radiation; $P$ the total precipitation (rain and snow); $T_a$ the air temperature; $Q_a$ the air humidity; $P_s$ the atmospheric pressure at the surface and $U$ the wind speed.

In the full version of SECHIBA-YAO, observations of LST or brightness temperature can be used to constrain model inner parameter or initial conditions of the model variables. However, the simulated LST is hemispheric and does not account for solar configuration and viewing angle effects. In order to compute a thermal infrared brightness temperature from LST, and neglecting the directional effects, the total energy emitted by the surface (Rad) can be computed using the following expression

$$Rad = k_{emis} \; \varepsilon \; LST^4 + \left(1 - \varepsilon \; k_{emis}\right) LW_{down} \qquad \text{(Eq 1)}$$

In this equation, $\varepsilon$ is the surface emissivity, $k_{emis}$ is the multiplicative factor for emissivity and $LW_{down}$ is the longwave incident radiation that is an input forcing of SECHIBA. Svendsen et al. (1990) proposed a transfer function to link the surface emitted radiance towards an observed brightness temperature $TB$ measured in the [8,14] spectral band The empirical formulation is given by the expression

$$TB = \left(\frac{Rad - 7.84}{6.7975.10^{11}}\right)^{0.2} \qquad \text{(Eq 2)}$$

In the following, the capabilities of the 4D-VAR is demonstrated in a series of assimilation experiment using the data provided by the FLUXNET network (Baldocchi et al., 2001). FLUXNET is a network coordinating regional and global analysis of observations from micrometeorological tower sites. The flux tower sites use eddy covariance methods (Aubinet et al. 2012) to measure the exchange of carbon dioxide ($CO_2$), water vapor, and energy between terrestrial ecosystems and the atmosphere. SECHIBA-YAO can be run using other data as long as the same inputs needed to operate SECHIBA are given

Measurement towers sprang up around the world, grouped in regional networks. The data from all networks is accessible to the scientific community via the Fluxnet website (http://www.fluxdata.org). In this work, we selected 2 sites: Harvard Forest and Skukuza Kruger National Park; both present contrasted climate and land surface properties suitable to test the tools developed and assess model parameters sensitivities. Only climate measurements with the same sampling frequency (30 minutes) from both sites are used to force SECHIBA. Vegetation characteristics are prescribed and only

homogeneous grids are considered. Two cases were studied with agricultural C3 (PFT 12) and bare soil (PFT 1).

*Skukuza Kruger National Park*

Located in South Africa at 25° 1' 11" S and 31° 29' 48" E , this Fluxnet site was established in 2000. The tower overlaps two distinct savanna types and collects information about land-atmosphere interactions. The climate is Subtropical-Mediterranean. The total mean annual precipitation is 650 mm, with an altitude of 150 m and the mean annual temperature is 22.15 ℃.

*Harvard Forest*

Located in the United States of America, on land owned by Harvard University, the station is located at 42º53'78" N and 72º17'15" W. It was established in 1991. The site has a Temperate-Continental climate with hot or warm summers and cold winters. The annual mean precipitation is 1071 mm, the mean annual temperature is 6.62 ℃ and the altitude is 340 m.

## 3. The Methodology

### 3.1 Variational assimilation

Variational assimilation (4D-VAR) (Le Dimet et al. 1986) considers a physical phenomenon described in space and its time evolution. It thus requires the knowledge of a direct dynamical model *M*, which describes the time evolution of the physical phenomenon. *M* computes geophysical variables, which are compared to observations. By varying some model parameters (control parameters), assimilation seeks to infer geophysical variables that are the closet to observation values (LST in the present case). The control parameters can be, as an example, initial conditions or physical parameters of *M* leading to the computation of LST.

The basic idea is to determine the minimum of a cost function *J* that measures the misfits between the observations and the model estimations. Due to the complexity of this function, the solution is classically obtained by using gradient methods, which implies the use of the adjoint model of *M*. This model is derived from the equations of the direct model *M*. The adjoint model estimates changes in the control variables in response to a disturbance of the output values calculated by *M*. It is done by integrating the same model in backward direction (e.g. time integration is from the future to the past). If observations are available, the adjoint allows minimizing the cost function *J*.

Formalism and notations for variational data assimilation are taken from Ide et al., (1997). *M* represents the direct model, $\mathbf{x}(t_0)$ is the initial state of the model and $\mathbf{k}$ represents the vector of the inner model parameters to be controlled, so $\mathbf{x}(t_i) = M_i(\mathbf{k}, \mathbf{x}(t_0))$, where $M_i(\mathbf{k}, \mathbf{x}(t_0))$ is represented by

$M \circ M \circ ... \circ M\big(\mathbf{k},\mathbf{x}(t_0)\big)$. The tangent linear model is noted $\mathbf{M}(t_i,t_{i+1})$, which is the Jacobean matrix of

$\mathbf{M}$, in $\mathbf{x}(t_i)$. The adjoint model $\mathbf{M}_i^T$ is the linear tangent transpose, defined as:

$$\mathbf{M}_i^T = \prod_{j=0}^{i-1}\mathbf{M}\big(t_j,t_{j+1}\big)^T \hspace{6cm} \text{Eq.(3)}$$

$\mathbf{M}$ is used to estimate variables, which are observed through an observation operator $\mathbf{H}$, permitting to

compare the observed values $\mathbf{y^0}$ with respect to the $\mathbf{y}$ calculated by the composition $\mathbf{H}\circ\mathbf{M}$, at the

location (in time and space) where observations are available.. We suppose that

$y_i = H_i\big(M_i(x_i,k)\big)+\varepsilon_i$ where $\bullet_i$ is a random variable with zero mean. This term represents the sum

of the model, observation and scaling error. Finally, the most general form of the cost function is

defined as follows:

$$J(\mathbf{k}) = \frac{1}{2}\big(\mathbf{k}-\mathbf{k}^b\big)^T \mathbf{B}^{-1}\big(\mathbf{k}-\mathbf{k}^b\big) + \frac{1}{2}\sum_{i=0}^{t}\big(\mathbf{y}_i-\mathbf{y}_i^0\big)^T \mathbf{R}_i^{-1}\big(\mathbf{y_i}-\mathbf{y}_i^0\big) \hspace{2cm} \text{Eq. (4)}$$

The background vector is defined as $\mathbf{k}^b$, which is an *a priori* vector of the inner model parameters.

The first part of the cost function represents the discrepancy to $\mathbf{k}^b$ and acts as a regularization term.

The second part represents the distance between the observations and the model estimates. $\mathbf{B}$ is the

covariance error matrix of $\mathbf{k}^b$ and $\mathbf{R}_i$ is the covariance error matrix of $\mathbf{y}^o$ at time $t_i$.

The objective of this work is to show the capacity of 4DVAR to help determining the value of the

principal inner parameters $\mathbf{k}$ of SECHIBA and the initial conditions for Surface Water Content. The

present distributed software allows the reader to do its own experiments using synthetic or actual

data. When the observations are synthetic (produced by the model itself) no transfer function from

the estimation to the observation are needed, and $\mathbf{H}$ is taken as the identity matrix. If actual data are

used, a specific $\mathbf{H}$ is used that transforms the soil temperature into brightness temperature (see

section Model and Data). In addition, the relationship prior value/actual value determines the

covariance matrix B, however in our case no covariance matrix is taken since the actual control

parameters values are out of the scope of this work. Finally, in our work, reading the covariance of

observations, the identity matrix is taken for $\boldsymbol{R}$.

The minimization of the cost function (Eq 4) is based on gradient-descent approaches. The cost

function gradient has the form

$$\nabla_k J = B^{-1}\big(k-k^b\big) + \sum_{i=1}^{t}M_i^T(k)\nabla_{yi}J \hspace{4cm} \text{Eq (5)}$$

Where $\nabla_k J$ and $\nabla_{yi}J$ are the gradients of the cost function $J$ with respect to $\mathbf{k}$ and $\mathbf{y}_i$ respectively.

The expression above allows us to compute $\nabla_k J$ by knowing $\nabla_{yi} J$, in the form of a matrix product of this term by the matrix $\mathbf{M}_i^T(\mathbf{x},\mathbf{k})$, corresponding to the transpose of the Jacobian Matrix. The development of calculation gives the expression of the gradient of **y**:

$$\nabla_k J = \mathbf{B}^{-1}(\mathbf{k} - \mathbf{k}^b) + \sum_{i=1}^{t} \mathbf{M}_i^T(\mathbf{k}) H^T \left[ R_i^{-1}(y_i - y_0) \right] \qquad \text{Eq (6)}$$

The control parameters are adjusted several times using a L-BFGS method (Gilbert and Lemarechal 1989) until a stopping criterion is reached.

## 3.2 YAO

Variational data assimilation requires the computation of the adjoint code of the direct model, which is a heavy and complex task, especially for a large model such as SECHIBA. Usually, the adjoint code is computed with the help of specific softwares (automatic differentiators) (e.g., Bischof et al.,1996; Giering and Kaminski, 2003; Hascoët and Pascual, 2004). These softwares are appropriate for the differentiation of large codes, but their use will be optimal only under specific coding conventions and a good level of modularity of the codes (Talagrand, 1991). Moreover, manual optimization of the produced code is often necessary. Therefore, in many practical cases the automatic production of code will not be totally optimal in terms of flexibility (e.g., when the direct model is updated frequently, one has to re-differentiate the whole code). These considerations motivated the development of a slightly different but complementary approach that focuses on the high-level structure of the numerical models, embedding implementation details inside simple entities that can be easily updated. This has led to the development of the YAO assimilation software at LOCEAN/IPSL (https://skyros.locean-ipsl.upmc.fr/~yao/).

YAO is based on the decomposition of a numerical model into elementary modules interconnected by directional links. On one side, the structure of the model (variables, dependencies...) is described as a graph structure. On the other side, the details of the physics are coded inside C/C++ basic modules that are ideally simple. The user can therefore separate the "high-level" structure of the model from implementation details. It is also very easy to update a numerical code within this framework. Regarding the assimilation strategy, YAO computes the tangent linear and adjoint codes from the elementary jacobians of each module (provided by the user). Adjoint/cost function test tools are also available. Finally, YAO includes routines devoted to classical assimilation scenario (incremental form ) and is interfaced with the M1QN3 minimizer (Gilbert and Lemaréchal, 1989). which has been designed to minimize functions depending on a very large number of variables, no subject to constraints. The algorithm implements a quasi-Newton technique (L-BFGS) with a dynamically updated scalar or diagonal preconditioner. It uses line-search to enforce global

convergence; more precisely, the step-size is determined by the Fletcher-Lemaréchal algorithm and realizes the Wolfe conditions.

**3.3 Graph formalism**

In YAO, a numerical model must be described as an ensemble of modules related by connections in order to form a graph. Let us define more precisely the main components of the graph:

- a *module* is a basic entity of computation, representing a deterministic (but possibly nonlinear) function transforming an input vector into an output vector. A module is viewed graphically as a node of the graph, the sizes of the vectors correspond to the number of input and output connections associated with the node.

- a *basic connection* is an oriented link relating two nodes of the graph. Most basic connections usually represent the transmission of the output of one module taken as input by another one.

The external context is the ensemble of data input and output points used as external data by a whole graph at a specific level of abstraction. Basic connections can link a data input point located in the external context to one or several module(s) (for instance modules needing the specification of some initial conditions, boundary conditions or model parameters). Inversely, the global outputs of the model link a module towards a data output point located in the external context.

The modular graph is the ensemble of the modules and of their connections. It must be acyclic so that a topological order may be defined on the nodes of the graph (i.e., if there is connection $F_p \rightarrow F_q$, then $F_p$ should be computed before $F_q$)   (see Fig.1)

Typically, a modular graph describes the equations governing the system of interest and each physical variable appearing in the governing equations are associated with a specific module. However, supplementary modules can also be defined to represent temporary variables required to simplify computations for complex equations. The user has generally to specify modules at a single point $(i, j, k, t)$ of space $(i, j, k)$ and time $(t)$, and the dependency to space-time locations (e.g. $i+1, j-1, k, t-1$) of the discretized variables taken as inputs. From the local description of the equations, YAO is able to build a model on a given space domain and on a given number of time steps by automatically replicating the local graph in space-time (cf. Fig.2)).

By passing the different modules in topological order, YAO is able to emulate the global model and to calculate the global model outputs given model initial conditions and parameters.

Now, we will see that the usefulness of the graph modular approach is reinforced when the jacobian matrix of each basic function is known. For a basic function F such that y = F( x ), the jacobian matrix F relates a perturbation of the inputs to the associated perturbation of outputs: $\mathbf{dy} = \mathbf{F}\,\mathbf{dx}$. Since the jacobian of a composition of functions is the product of the elementary jacobians, the tangent linear model associated with a modular graph may also be obtained by passing the graph in the same topological order.

The "lin-forward" algorithm is the following:

1) Initialize the external context data input points with a perturbation $\mathbf{dx}_i$ (around a given linearization point)

2) Pass the modules in topological order and propagate the perturbation

3) Estimate the perturbation output $\mathbf{dy}$ on output data points in the external context of the graph.

Following this procedure, YAO can emulate the global tangent-linear model from elementary jacobians. In the same manner, a backward algorithm may be defined for adjoint computations. From (Eq. 1), it may be shown that the global adjoint will be retrieved by back-propagating the graph, with a few adjustments not detailed here (see, Nardi et al., 2009 for more details on the "backward" algorithm). This property is the basis of the semi-automatic adjoint computation by YAO.

An implementation of a variational assimilation procedure with YAO follows the structure represented in Fig. 3. The YAO compiler builds an executable file following the scheme presented in Fig.3. This file is independent of the assimilation instructions. The executable file reads these instructions from an instruction file. Due to the graph structure of the model and of its adjoint, it is easy to modify the model and its adjoint, e.g. by updating some adequate modules; one can systematically obtain the update global direct model and the global adjoint

As mentioned in the introduction, this paper gives access to a compiled version of SECHIBA-YAO and allows to perform some assimilation experiments related to the control of the ten most influent internal parameters of SECHIBA by observing the land surface temperature .  YAO is a free software that gives the opportunity to modify the SECHIBA code provided in this paper.

## 3.4 Development of SECHIBA-YAO

The implementation of SECHIBA in YAO starts with the definition of the modular graph describing the dynamics of the model (see Appendix A). Elementary processes and interconnections between modules are defined in order to represent the computation flow in the model. -The modular graph was built as follows:

-Every component of the original code was carefully studied line by line directly.

-A list of inputs and outputs for each subroutine was made, for every routine of SECHIBA. This permits to exactly know the information flow in the model.

-A second zoom in the subroutines was made in order to understand the internal dynamics of the code. This is the last step in the modular graph definition. When studying the subroutines, their complexity was reduced by breaking the different steps into simpler elements. The idea is to have a scalable code. Uncoupled modules give more independence when changing part of the model. Cohesive modules help to understand the model.

-The original six subroutines in the SECHIBA-Fortran code are split into 130 modules  by the SECHIBA-YAO modular graph, corresponding to every process modeled by SECHIBA and to a number of transitional modules serving as auxiliary computing.

-It is important to mention that every variable and subroutine name was kept as in the original model. If a user or developer of SECHIBA-Fortran sees the implementation in YAO, he will find his way easily.

### 3.4.1 Direct model

After defining the modular graph in YAO, the second step in the SECHIBA-YAO implementation is the coding of the direct and the derivatives of the modules.. Every module is represented as a source file and the different processes attributed to the module are implemented inside the source file, allowing a better control of the physics, i.e. any change in the physics could be made easily.

### 3.4.2 Module Derivatives

Once the direct model has been coded and validated, there are two options to code the derivatives: they can be coded line-by-line based on the forward computing, in order to obtain the jacobian matrix of the module, or they can also be produced routinely, using an automatic differentiation tool (for example, Tapenade (Hascoët et al, 2012)). For SECHIBA-YAO, the derivative process was made line-by-line. The outputs are derived with respect to every input. YAO generates automatically, based on these derivatives, the tangent linear and the adjoint of the model.

Nevertheless, the derivative process introduced errors related to the coding process, to inexact derivatives (e.g. expressions that were not differentiable). In order to reduce it to a minimum number of bugs, the adjoint of the model was validated (as it was made with the direct model). This guarantees the accuracy when performing assimilation. The validation of the adjoint model is presented in section 4.1

## 4. Data assimilation experiments

In this section we present several experiments that have been realized using the SECHIBA-YAO system. They were designed to control the eleven most influent internal parameters of SECHIBA when we assimilate the land surface temperature LST.

In order to deal with non-dimensional control parameters with the same order of magnitude, a preprocessing has been applied. The control parameters were first divided into two groups. The first group includes physical parameters, which have a physical dimension. In the present work, these parameters were normalized by dividing them with their prior values in order to control non-dimensional parameters. In such a way, given that the prior value is the true value (in the case of twin experiments), a value of one for these parameters indicates that the control parameter has been correctly reconstructed. The second group corresponds to physical parameters that are multiplied by a 'multiplicative factor' (Table 1), which is dimensionless (Verbeeck et al, 2011). The multiplying factors are the control variables of the second group and are set to one at the beginning of the assimilation process. The normalization process on the one hand and the use of multiplicative factors on the second hand allow us to deal with numbers of the same order of magnitude, which facilitates the comparison of the sensitivity of the different control variables in the assimilation process.

In the following, all variable are supposed to be preprocessed, so they are normalized and centered around one.

The model inner parameters are the following (see Table 1): $rsol_{cste}$ is a numerical constant involved in the soil resistance to evaporation. This parameter limits the soil evaporation, so the greater its value the lower the evaporation; $hum_{cste}$, $mx_{eau}$ and $min_{drain}$ are related to soil water processes, the higher their values, the more water will be available in the model reservoir, affecting water transfers and especially evapotranspiration; $dpu_{cste}$ represents the soil depth in meters. The other parameters are multiplicative factors: $k_{rveg}$ which is used in the calculation of the stomatal resistance, this variable limits the transpiration capacity of leaves, the greater its value, the lower the transpiration; $k_{emis}$ controls the soil emissivity used to compute land surface temperature. This parameter takes part in the net radiation calculation which determines the energy balance between incoming and outgoing surface fluxes; $k_{albedo}$ weights the surface albedo, which is defined as the reflection coefficient for short wave radiation; $k_{cond}$ and $k_{capa}$ take part in the thermal soil capacity and conductivity, both involved in the computation of the soil thermodynamics and $k_{z0}$ weights the roughness height, which determines the surface turbulent fluxes. Since the control parameters are normalized, we apply a perturbation which is of the form of a random noise limited up to 50% of the true parameter whose value is one, so the perturbed value belongs to [0.5 , 1.5]. If the control parameter values posterior to

1 the assimilation process are close to 1, it means that the assimilation was successfully achieved.

Differences between the values retrieved and the prior values represent relative errors on the

parameter estimation, posterior to assimilation.

In order to show the benefit of data assimilation in SECHIBA, we conducted several experiments

using SECHIBA-YAO. Prior to the assimilation process, different scenarios were defined for the

tests (Table 3). A scenario makes reference to the experimental conditions. It includes the definition

of the vegetation functioning type (PFT), the type of observation to be assimilated, the observation

sampling, the time sampling, the atmospheric forcing file, the subset of control parameters, the

assimilation window size and the time of the year to start the assimilation. The different scenarios

were calculated using the adjoint model for several typical conditions of the two Fluxnet sites

selected. The dates presented in this paper are representative of sunny days in summer or winter,

with no perturbation coming from clouds and without rainfall events. In equation 4, we take R as the

identity matrix that means that we assume the errors of the observations are uncorrelated. The next

section explains the scenarios for the different experiments performed in this work.

**4.1 Variational sensitivity analysis**

In order to show the accuracy of the distributed SECHIBA-YAO code, we present an analysis that

allows to rank the eleven parameters according to their sensibility estimated by using the adjoint

model and to compare the results to those obtained by using finite differences. We identify the most

sensitive parameters to the estimation of land surface temperature *(LST)* by computing the gradients

obtained with the adjoint model. This analysis corresponds to a first-order sensitivity estimate of the

influence of the control parameters on the land surface temperature. In order to do so, local

sensitivities were determined by computing the parameter gradients both by finite difference and by

adjoint calculation (Saltelli et al, 2008). This method is really local and the information provided is

related to a definite point in space. The values of the 11 parameters concerned in the analysis are

presented in Table 1, they represent the initial values where the experiments have been conducted.

Because $hum_{cste}$ is related to vegetation type, in this work only value for PFT 1 (5 m-1) and PFT 12

(2 m-1) are considered.

The sensitivity analysis was performed for a subset of inner parameters related to the energy and

water physical processes on bare soil (PFT 1) and agricultural C3 crop (PFT 12), in order to quantify

the role of the vegetation on the land surface temperature parameters' sensitivity. The land functional

types are useful to distinguish the different soil type. In the present case we used the agricultural C3

grass type whose parameters are: *Vcmax*, *opt* (optimal maximum rubisco-limited potential

photosynthetic capacity) = 90 $mol/m^{-2}s^{-1}$; *Topt* (Optimum photosyntixc temperature) = *27,5+0,25Tl* *°C*; *Tl* (Function of multiannual mean temperature for C3 grasses); *maxLAI* (maximum LAI beyond whitch there is no allocation of biomass to leave) = 6 ;*zroot* (exponential depth scale for root length profile) = *0,25m*; *leaf* (prescribed leaf albedo) = *0,18*; *h* (prescribed height of vegetation) = *0,4m; Ac* (Critical leaf senescence) = *150 days*; *Ts* (weekly temperature beyond which leaves are shed if seasonal temperature trend is negative) =*10°C*; *Hs* (weekly moisture stress beyond which leaves are shed) = *0,2.*

The work was made on a daily basis, in order to observe the diurnal variations of sensitivities. At each half-hour time step, model outputs are computed. At each time step, a gradient is computed in order to have the updated gradient value. As we make the assumption that the error on prior values are very large in comparison with error on observations, we discard the background term in the cost function (defined in section 2). This simplification is valid as soon as the system is overdeterminated (i.e. the number of control parameters is smaller than the number of observatios). The initial values of the parameters (before optimization) are those of Table 1. We recall that for numerical purpose, the control parameters have been normalized in order to have the same order of magnitude (i.e. equal to 1). Calculations were performed for both FLUXNET sites considered in this work.

Figure 4 compares, for August 28,1996 at Harvard Forest, the sensitivities computed for each control parameter with both finite differences and model gradients. Bare soil results are presented in Fig.4(a). The agricultural C3 crop scenario is illustrated in Fig.4(b). The efficiency of the adjoint calculation is first demonstrated in these plots, because the 11 desired parameter sensitivities are obtained in a single integration, whereas it takes 11 runs of the model to compute the same quantity using finite differences. By using the same methodology, sensitivity curves were computed in the FLUXNET site Kruger Park (Figure 5). The comparison between sensitivity analysis done using the adjoint and using finite differences shows a very good agreement between the two methods for both sites. The diurnal characteristics of the parameter sensitivities with a maximum around noon in phase with the diurnal variation of solar radiation are clearly visible.

Table 2 presents, for Harvard Forest and Kruger Park, the 11 parameters ranked with respect to their influence. According to the four scenarios defined (two sites and two PFT), it can be seen that the hierarchy changes with the vegetation, but remains the same for both sites. Parameter hierarchy revealed that the highest gradient values correspond to those that have the largest influence on the land surface temperature estimate. Clearly $k_{emis}$ is the most influential parameter in the calculation of land surface temperature, regardless of the climatology used and vegetation fraction. In addition, $min_{drain}$ is the least influential parameter for all scenarios.

The parameters $k_{capa}$, $k_{cond}$, $k_{zo}$ and $k_{albedo}$ are the most influential in bare soil conditions, after $k_{emis}$. In the presence of vegetation, several sensitivities change radically: $k_{rveg}$ becomes the most important multiplicative factor after $k_{emis}$; the factor $k_{albedo}$ is less sensitive compared to its influence in the bare soil case and $mx_{eau}$ is more sensitive, given that less water is available when a fraction of vegetation is present. The other parameters show equivalent sensitivity values regardless the scenario. For $hum_{cste}$ and $k_{rveg}$, sensitivities are equal to zero for bare soil, because these parameters affect surface temperature only in presence of vegetation.

Parameters with persistent positive sensitivity are: $rsol_{cste}$, $k_{rveg}$ and $hum_{cste}$ . Parameters with persistent negative sensitivity are: $k_{z0}$, $k_{albedo}$ and $emis$. The sign of the gradients reflects the positive or negative feedback on the surface temperature of the processes involved. For example, the parameters involved in the evapotranspiration processes present negative sensitivities because a reduction (respectively an increase) of the evapotranspiration will lead to an increase (respectively a decrease) of the land surface temperature, when the soil water content is sufficient.

Transpiration processes influence directly the land surface temperature in presence of vegetation and is the dominant process in the studied sites. Therefore $k_{rveg}$ has a higher sensitivity than $k_{cond}$, $k_{capa}$ and $k_{albedo.}$ . For bare soil, on the contrary, the dominant processes are those related to the soil thermodynamics, explaining why $k_{capa,}$ $k_{cond}$ and $k_{emis}$ are the most sensitive parameters.

In general, sensitivities are higher in bare soil conditions for the control parameters, except for $min_{drain}$ and $mx_{eau}$. Since $min_{drain}$ is not sensitive to the land surface temperature, this parameter is no longer controlled. Only the ten most influent parameters are used in the following sections.

The next section presents the different assimilation experiments that we have performed using the SECHIBA-YAO software.

## 4.2 Twin experiments

Twin experiments permit to check the robustness of the variational assimilation method by assimilating synthetic data. First the direct model is run with a set of parameters *Ptrue* (the initial conditions) in order to produce pseudo observations of land surface temperature *LST*. Then *Ptrue* is randomly noised to obtain *Pnoise*. Assimilations of land surface temperature *LST* were then performed in the model run with *Pnoise* as new initial conditions for the control parameters during several days (most of the time, one week), leading to a new set of optimized parameters denoted *Passim*. *Passim* is then compared to *Ptrue* in order to estimate the performances of the assimilation process. Five different assimilation experiments were performed. These experiments are available in the distributed version of SECHIBA-YAO.

**4.3 Definition of experiments**

The 10 most sensitive parameters are considered in the twin experiments (all the above parameters except $min_{drain}$). We present hereinafter the results obtained with different assimilation realizations. Each assimilation experiment was conducted by perturbing the initial conditions of the control parameters with a uniform distribution random noise reaching 50% of the parameter nominal values. This procedure permitted us to obtain the relative errors of the control parameters and the root mean square error (RMSE) of the model fluxes, based on their value before and after the assimilation process.

A scenario for a single experiment is defined by several properties, described in Table 3. Scenarios for all the assimilation experiments are presented in Table 4. All parameters are controlled at the same time. The duration of each assimilation experiment is one week or one month depending on the experiment. The time steps $\Delta T$ of each experiment is 30 minutes except for experiment 1 where the time step varies. All experiments presented in this work use Harvard Forest and Kruger Park as forcing. For each experimental setting, 5 different assimilation realizations were made except for experiment 2 where 500 independent assimilation where run. The mean errors are presented in table 4.

In experiments 1 and 2, the six most sensitive parameters are controlled. In both cases the vegetation type is PFT 12. In Experiment 1 several observation assimilation samplings are tested going from 30 minutes up to 24 hours. During one month, five independent assimilation tests were run for each observation sampling. In Experiment 2, a weighted random noise was introduced in the observations, going from 10% up to 50% of the true value of the observation. Both Experiments 1 and 2 use constant perturbations of the control parameters (50% of its prior value for experiment 1 and 10% for experiment 2) in order to assess the impact of varying the observation sampling and the noise in the observations.

In Experiment 3 the five most sensitive parameters according to the sensitivity analysis (Table 2) were controlled in bare soil conditions (PFT 1) in Harvard Forest and Kruger Park sites. In this experiment the noise added on the prior values is 50%.

In Experiment 4 the five most sensitive parameters for each PFT were controlled in the conditions of agricultural C3 (PFT 12), according to the sensitivity analysis (Table 2), in Harvard Forest and Kruger Park sites. Doing so, we were able to assess the effect of the vegetation fraction on the assimilation system. In addition, taking only the most sensitive parameters in the control set permitted to increase the assimilation performances, given that the more the observed variable is sensitive to a parameter, the easier the minimization process finds its optimal value, and

consequently reducing the estimation error. In this experiment the noise added on the prior values is 50%.

In Experiment 5, all parameters, except $min_{drain}$, were controlled (since $min_{drain}$ has no impact in the land surface temperature estimation), during a week in Harvard Forest and Kruger Park.

Comparing Experiment 5 with Experiments 3 and 4 allows us to study the impact of taking a larger number of control parameters in the assimilation process. In addition, we want to test if LST observation provides enough information to constrain all the model parameters at the same time and if we can hope to improve all model state variables. In this experiment the noise added on the prior values is 50%.

## 4.4 Results

### 4.4.1 Effect of the observation sampling

Experiment 1 investigates the impact of the observation sampling (2mn, 2h, 6h, 12h, 24h) in the assimilation, since varying the observation frequency leads to vary the amount of observations available. Each test was labeled with a number. This number serves as a reference to compare the different results. Table 5 presents the several tests we conducted as well as their initial conditions. For example, in Test 4, only two observations per day are taken at noon and at midnight. In Test 5, we have one observation per day, taken at noon, and so on.

Prior and final errors before and posterior to the assimilation process are presented in Table 6 for Kruger Park and Harvard Forest sites. The columns represent the different assimilations performed with different frequency sampling in the observations. Five independent assimilations were made for each test. Table 6 reports the mean value of the performances of the assimilation system. Even though small errors were found for the different tests, we do notice that the assimilation system is sensitive to the observation sampling.

The contribution of the observations is demonstrated by an improvement in the optimization when increasing the frequency of observations, both for the controlled parameters and the computed fluxes $H$ and $LE$ that are major outputs of the model. The final error values in the different tests increase by a factor of 10 when reducing the sampling frequency.

### 4.4.2 Effect of random noise in the observation

Experiment 2 aims at studying the impact of introducing a random noise in the synthetic observations. The random noise follows a normal distribution with zero mean and variance 1. The perturbed observations are computed using the following equation

$$LST^* = LST + amp \cdot \phi \qquad \text{Eq (7)}$$

with *LST\** the perturbed observation, *LST* the original land surface temperature, *amp* a factor weighting the random noise going from 10% to 50%, and $\phi$ the normal distribution random noise. The control parameter set is composed of the six most influential parameters in the computation of LST. The initial conditions of the parameters are obtained by perturbing them 10% uniformly from their prior values. Three tests were performed, aiming to check the impact of introducing different magnitudes of errors prior the assimilation process. Results are presented in Table 7. Mean value of the five hundred independent assimilations is presented. Posterior to each experiment, the parameter relative error and the model flux RMSE are computed to quantify the quality of the results.

We note in Tables 7.(a) and 7.(b) that the parameter restitution is degraded when adding random noise to the observations. This shows that the sensitivity of the assimilation system is quite sensible to the noise affecting the LST observations. When increasing the amplitude of the error, the various errors obtained for the three tests not only suggest the need to take into account the quality of the observations in the model but also the fact that the parameters are not affected in the same way by the data uncertainties. However, perturbations are still limited and a deeper exploration should be performed to assess impact in the assimilation performance of noisy observations.

### 4.4.3 Effect of the control parameter set size

The RMSE errors of the assimilations for experiments 3,4 and 5 are presented in Table 8 9, corresponding to Harvard Forest and Kruger Park sites. For all the experiments the noise added on the parameters was 50%.

In experiment 3 for PFT 1, the mean errors on the retrieved values for all the control parameters are of the order of $10^{-8}$. Regarding the LST retrieving, the mean RMSE ranges from 4.82 K prior assimilation to $2.1.10^{-5}$ K after the assimilation process. The same behavior is observed for the different model fluxes. Both FLUXNET sites used as forcing have more or less the same behavior, in regards of the error reduction.

In experiment 4 for PFT 12 similar results were observed. The assimilation process permits the reduction of the parameter errors for both sites and both PFT used (Table 8.b).

In experiment 5, the relative value of the RMSE with respect to the synthetic measurements, for *LST*, *LE* and *H* is reduced at both FLUXNET sites. The same results hold for the mean relative error of the control parameters.

Comparing the results from Experiments 3 and 4 to Experiment 5, degradation in fluxes and parameter restitution can be observed. Effectively, we find higher errors in the fluxes and the final control parameters when increasing the size of the control parameter set (Experiment 5). The best

performances in the parameters restitution are obtained when controlling 5 parameters only. When we control the 10 most sensitive parameters, as in Experiment 5, degradation in the final value of the parameters is observed. Indeed, the larger the control parameters set, the more easily the cost function may converge toward a local minimum (that can be far from the global optimum). In addition, it is difficult to retrieve accurately parameters that are insensitive to LST, thus the assimilation of this variable in order to optimize these parameters is not efficient.

## 5. Discussion and Conclusion

In this study the adjoint of SECHIBA was implemented, using an adjoint semi-generator software denoted YAO. The land surface temperature gradients with respect to each control parameter were computed by SECHIBA-YAO, which permitted us to carry out a sensitivity analysis of the parameter influence on the synthetic LST estimation on the one hand and to conduct several assimilation experiments on the other hand.

The first contribution of this paper was the sensitivity analysis results. They showed exactly which parameters of the model are the most sensitive and have to be controlled during the assimilation process. However, it is important to mention that sensitivity analysis depends on the region, the forcing, the PFT, the time period (hour and day), among other factors. Once the parameter hierarchy was set, twin experiments were performed for different scenarios, aiming at testing the robustness of the assimilation scheme.

The second contribution of this work is that we showed the usefulness of the variational data assimilation of LST (Land Surface Temperature) to improve the SECHIBA parameter estimations. LST assimilation has the potential of improving the LSM parameter calibration, by adjusting them properly during the control process. In a forecasting approach, this can be valuable, due to the fact that the simulation can be more reliable since the model parameters are fitted on actual measurements. Improvement in the fluxes computed by the model after the assimilation of LST was demonstrated. Twin experiments showed the power of variational data assimilation to improve the model parameter estimation. Different experiments conducted for different scenarios and forcing sites were successfully accomplished, meaning that a reduction in the fluxes errors was obtained by introducing information given by the LST synthetic observations. In addition, the influence of the size of the control parameter set in the assimilation performance was proven.

Estimating only the most sensitive parameters to LST increases our chances to find acceptable values for them after assimilation. Optimizing a larger control parameter set, as in Experiment 5, makes more difficult for the assimilation system to retrieve the prior value of the control parameters with a high accuracy. After presenting the different experiments, some aspects of data assimilation arise

when analyzing the results. The first one concerns the presence of several local minima due to the non-linearity of the SECHIBA model. Second, we have also shown a significant improvement in the assimilation performances when the sampling frequency of observations is increased, as evaluated in Experiment 1. This suggests that the ability of the model to be constrained depends, among other things, on the observation frequency. By decreasing the number of observations, the control parameter adjustment is less accurate, and the assimilation procedure estimates variables with a larger error. Therefore it can be verified that if we have more LST observations, the assimilation system will fit the parameters better, so improved estimations are obtained.

Finally, we observe a strong dependence between the quality of observations and the parameter restitution, as shown in Experiment 2. It seems crucial to take into account the uncertainty in the observations, because they do not affect the assimilation performance in the same way when estimating each parameter in the minimization process. If we compare experiments 1 and 2 (Table 6 and 7), it is clear that the noise on the observations dramatically increases the mean error on the computed fluxes L and H; the LST that are assimilated, are retrieved with a better accuracy. The introduction of a regularization term on the parameters could be used to mitigate this problem. Constraining parameters and weighting observations according to their confidence in the minimization phase can be modeled through the introduction in the cost function of ~~the~~ variance-covariance errors matrices (background B and observation R). It is an important aspect to consider for assimilating real observations.

Adding extra parameters to the control set increases the complexity of the cost function. By taking into consideration the results of assimilation of LST when controlling the 10 most sensitive parameters (Experiment 5), we could see that, after having made several assimilation runs, LST does not provide enough information to constrain the parameter set, in order to improve the estimation of the SECHIBA parameters. In the case of controlling all parameters we cannot hope improving the estimation of all model parameters unless we assimilate additional observations or we add a background term in the cost function.

Assimilation with the YAO approach permits the implementation of different assimilation scenarios in a very flexible way, when performing different twin experiments: the control parameters and the observed variables (once the adjoint code has been generated), the assimilation windows, the observation sampling, the time sampling and other different features can be changed easily.

A distributed version of SECHIBA-YAO code and several examples with different scenarios are available at a GitHub dedicated site. YAO can be downloaded upon request at https://skyros.locean-ipsl.upmc.fr/~yao/. Direct use of this software will allow performing other experiments using different physical conditions or even changing several equations of the model.

## 6. Code and data availability

The distributed version of SECHIBA-YAO provides an opportunity for scientists to perform their own assimilation. The distributed version allows the control of the 5 most influent internal parameters of SECHIBA, depending on the vegetation type. In addition, LST or satellite brightness temperature can be used as observations.

The distributed version of SECHIBA-YAO is available in a GitHub repository (https://github.com/brajard/sechibavar/archive/v1.0.zip), the user can download the software, save it in a local repertory and run the *makefile* in order to build a local executable. Documentation and two instruction files are available in order to guide the user towards their own implementation. Users can modify the forcing file, the initial date to the assimilation, the parameters value and their perturbation if needed. The assimilation frame (1 week), the step time (30 minutes), the observed variable (land surface temperature), the control parameters (only 5) and other initial parameters are imposed. If user wants to have access to a full modifiable version, YAO software has to be installed (https://skyros.locean-ipsl.upmc.fr/~yao/ ).

The instructions files given with the distributed version correspond to the twin experiments presented in this paper. Initial parameters like the assimilation time frame and the observed variable (LST) cannot be changed in the distributed version. However the other initial parameters used to build different scenarios can be changed easily through the instruction file (initial parameter values, PFT, observations files, forcing, initial date, etc).

## Acknowledgements

This work used eddy covariance data acquired by the FLUXNET community and in particular by the following networks: AmeriFlux (U.S. Department of Energy, Biological and Environmental Research, Terrestrial Carbon Program and AfriFlux) and the global FLUXNET project (http://daac.ornl.gov/FLUXNET/fluxnet.html). A special thanks to M. Crepon from LOCEAN for his active participation in the revision of this article. Dr. P. Peylin and F. Chevalier are acknowledged for fruitful discussions. We thank also Dr. F. Maignan for its continuous support in the use of ORCHIDEE model, and Dr. M. Berrondo, for the assistance in writing this article.

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

| Parameter | Description | Prior Value | Unit |
|-----------|-------------|:-----------:|:----:|
| **Inner Parameters** | | | |
| $rsol_{cste}$ | Evaporation resistance | 33000 | $S/m^2$ |
| $hum_{cste}$ | Water stress | {5, 2} | $m^{-1}$ |
| $mx_{eau}$ | Maximum water content | 150 | $Kg/m^3$ |
| $min_{drain}$ | Diffusion between reservoirs | 0,001 | $S/m^2$ |
| $dpu_{cste}$ | Total depth of soil water pool | 2 | m |
| **Multiplying Factors** | | | |
| $k_{emis}$ | Surface Emissivity | 1 | - |
| $k_{rveg}$ | Vegetation Resistant | 1 | - |
| $k_{albedo}$ | Surface albedo | 1 | - |
| $k_{cond}$ | Soil Conductivity | 1 | - |
| $k_{capa}$ | Soil Capacity | 1 | - |
| $k_{z0}$ | Roughness height | 1 | - |

Table 1. SECHIBA control parameters used in this work. There are 5 inner parameters, involved in the model estimations and 6 multiplying factors that are imposed to specific fluxes

| Site | Bare Soil (PFT 1) | Agricultural C3 crop (PFT 12) |
|---|---|---|
| Harvard Forest | $k_{emis}$, $k_{cond}$, $k_{capa}$, $k_{z0}$, $k_{albedo}$, $dpu_{cste}$, $rsol_{cste}$, $mx_{eau}$ $min_{drain}$, $k_{rveg}$ $hum_{cste}$, | $k_{emis}$, $k_{rveg}$, $k_{cond}$, $k_{capa}$, $k_{z0}$, $mx_{eau}$, $hum_{cste}$, $k_{albedo}$, $dpu_{cste}$, $rsol_{cste}$ $min_{drain}$ |
| Kruger Park | $k_{emis}$, $k_{cond}$, $k_{capa}$, $k_{z0}$, $k_{albedo}$, $dpu_{cste}$, $rsol_{cste}$, $mx_{eau}$ $min_{drain}$, $k_{rveg}$ $hum_{cste}$, | $k_{emis}$, $k_{rveg}$, $k_{cond}$, $k_{capa}$, $k_{z0}$, $mx_{eau}$, $hum_{cste}$, $k_{albedo}$, $dpu_{cste}$, $rsol_{cste}$ $min_{drain}$ |

3
4 Table 2. Sensitivity analysis results. Parameter hierarchy according to each site and vegetation
5 fraction. The parameters are ranked by decreasing sensibility.

| Properties | Description |
|---|---|
| Assimilation period | Time window of the assimilation period. |
| Number of assimilations | For each experiment, a number of assimilation is made with the same scenario but with different control parameter initial values |
| Control Parameters | Parameters to be optimized in the assimilation procedure. The number of parameters depend on the experiment. They are chosen among the following parameters: $k_{emis}$, $k_{cond}$, $k_{capa}$, $k_{z0}$, $k_{albedo}$, $dpu_{cste}$, $rsol_{cste}$, $mx_{eau}$ $min_{drain}$, $k_{rveg}$ $hum_{cste}$, |
| Observations | Model variables considered as observations: $LST$ in the present study |
| Observation sampling | Frequency sampling of the observations |
| Forcing | Data forcing used to perform the assimilation for a given site and a given date |
| Vegetation type | Vegetation fraction considered in the experiment |

2
3    Table 3. Scenarios properties and description

| Scenario | Experiment 1 | Experiment 2 | Experiment 3 | Experiment 4 | Experiment 5 |
|---|---|---|---|---|---|
| **Assimilation period** | 11/02/2003, 1 month Kruger Park 28/08/1996, 1 month Harvard Forest | 11/02/2003, 1 week (Kruger Park) 28/08/1996, 1 week (Harvard Forest) | 11/02/2003, 1 week (Kruger Park) 28/08/1996, 1 week (Harvard Forest) | 11/02/2003, 1 week (Kruger Park) 28/08/1996, 1 week (Harvard Forest) | 11/02/2003, 1 week (Kruger Park) 28/08/1996, 1 week (Harvard Forest) |
| **Number of assimilations** | 5 experiments for each site | 500 experiments for each site | 5 experiments for each site | 5 experiments for each site | 5 experiments for each site |
| **Control Parameters** | $k_{rveg}$, $k_{emis}$, $k_{cond}$, $k_{capa}$, $k_{z0}$, $k_{albedo}$ | $k_{rveg}$, $k_{emis}$, $k_{cond}$, $k_{capa}$, $k_{z0}$, $k_{albedo}$ | $k_{emis}$, $k_{cond}$, $k_{capa}$, $k_{z0}$, $k_{albedo}$ | $k_{emis}$, $k_{rveg}$, $k_{cond}$, $k_{capa}$, $k_{z0}$ | All parameters, except $min_{drain}$ |
| **Observations** | Soil Temperature | Soil Temperature with noise | Soil Temperature | Soil Temperature | Soil Temperature |
| **Observation sampling** | 30 minutes, 2, 6, 12 and 24 hours | 1 hour | 30 minutes | 30 minutes | 30 minutes |
| **Forcing** | Kruger Park and Harvard Forest | Kruger Park and Harvard Forest | Kruger Park and Harvard Forest | Kruger Park and Harvard Forest | Kruger Park and Harvard Forest |
| **Vegetation type** | PFT 12 | PFT 12 | PFT 1 | PFT 12 | PFT 12 |

4
5
6 Table 4. Characteristics of the scenarios for each of the twin experiments

| Test Number | Sampling Frequencies | Observations per day | Observation per month |
|---|---|---|---|
| 1 | 30 minutes | 48 | 1440 |
| 2 | 2 hours | 24 | 720 |
| 3 | 6 hours | 4 | 120 |
| 4 | 12 hours | 2 | 60 |
| 5 | 24 hours | 1 | 30 |

3

Table 5. Sampling frequencies for Experiment 1

|  | Fluxes | Prior | **RMSE** | | | | |
|---|---|---|---|---|---|---|---|
|  |  |  | **1** | **2** | **3** | **4** | **5** |
| Kruger Park | $H$ (W/m²) | 25 | 0.437 | 0.138 | 0.43 | 4.7 | 10.34 |
|  | $LE$ (W/m²) | 15.7 | 0.0601 | 0.592 | 0.594 | 2.43 | 10.8 |
|  | $LST$ (K) | 7.98 | 0.0601 | 0.0243 | 0.592 | 0.594 | 1.9 |
| Harvard forest | $H$ (W/m²) | 13.42 | 0.15 | 0.98 | 1.84 | 3.98 | 4.08 |
|  | $LE$ (W/m²) | 86.23 | 0.22 | 0.35 | 3.81 | 5.17 | 11.95 |
|  | $LST$ (K) | 5.98 | 0.08 | 0.65 | 0.86 | 1.27 | 1.61 |

(a)

|  | Control Parameters | Noise Interval in% | **Error (in %)** | | | | |
|---|---|---|---|---|---|---|---|
|  |  |  | **1** | **2** | **3** | **4** | **5** |
| Kruger Park | $k_{cond}$ | 50 | 0.0183 | 0.261 | 0.340 | 0.921 | 4.96 |
|  | $k_{capa}$ | 50 | 0.0427 | 0.172 | 0.4006 | 0.91 | 3.77 |
|  | $k_{z0}$ | 50 | 0.00103 | 0.0162 | 0.147 | 0.24 | 1.34 |
|  | $k_{rveg}$ | 50 | 0.418 | 0.909 | 3.845 | 4.01 | 14.97 |
|  | $k_{emis}$ | 50 | 0.1704 | 0.2733 | 0.77 | 1.27 | 4.4 |
|  | $k_{albedo}$ | 50 | 0.128 | 1.384 | 3.214 | 4.15 | 25.01 |
| Harvard forest | $k_{cond}$ | 50 | 0.37 | 0.54 | 3.7 | 5.7 | 10.14 |
|  | $k_{capa}$ | 50 | 0.36 | 2.86 | 4.16 | 10.55 | 20.74 |
|  | $k_{z0}$ | 50 | 0.0592 | 0.15 | 7.61 | 13.74 | 16.73 |
|  | $k_{rveg}$ | 50 | 0.31 | 0.75 | 5.25 | 7.24 | 17.8 |
|  | $k_{emis}$ | 50 | 0.11 | 0.17 | 5.82 | 10.86 | 13.74 |
|  | $k_{albedo}$ | 50 | 1.54 | 4.81 | 12.69 | 34.11 | 37.8 |

(b)

Table 6. Results of Experiment 1 using Harvard Forest and Kruger Park sites. (a) The first two columns give the computed fluxes prior the assimilation. The last five columns present the RMSE (prior-estimated) for each run
(b) The first two columns give the noise interval (in %) introduced for each control parameter with respect to the initial value of 1. The last five columns present the Relative error in % on the control parameters for the five sampling frequencies reported in table 5.

|  | | | | RMSE | |
| --- | --- | --- | --- | --- | --- |
|  | **Fluxes** | **Prior** | **10%** | **30%** | **50%** |
| Kruger Park | $H\,(W/m^2)$ | 24.7 | 7.26 | 7.81 | 8.32 |
|  | $LE\,(W/m^2)$ | 4.06 | 3.78 | 3.9 | 6.22 |
|  | $LST\,(K)$ | 7.12 | 0.019 | 4.48 | 6.23 |
| Harvard Forest | $H\,(W/m^2)$ | 25 | 5.92 | 11.13 | 24.01 |
|  | $LE\,(W/m^2)$ | 15.7 | 4.77 | 14.04 | 15.05 |
|  | $LST\,(K)$ | 7.98 | 0.046 | 1.42 | 2.59 |

(a)

|  | | | Mean error (%) | | |
| --- | --- | --- | --- | --- | --- |
|  | **Control Parameters** | **Noise interval in %** | **10%** | **30%** | **50%** |
| Kruger Park | $k_{cond}$ | 10 | 4.5 | 4.9 | 11.12 |
|  | $k_{capa}$ | 10 | 1.51 | 3.35 | 14.9 |
|  | $k_{z0}$ | 10 | 3.24 | 3.99 | 4.09 |
|  | $k_{rveg}$ | 10 | 6.91 | 10.1 | 11.5 |
|  | $k_{emis}$ | 10 | 2.79 | 3.14 | 4.08 |
|  | $k_{albedo}$ | 10 | 1.12 | 2.01 | 3.02 |
| Harvard Forest | $k_{cond}$ | 10 | 0.83 | 4.32 | 7.6 |
|  | $k_{capa}$ | 10 | 4.47 | 9.05 | 9.21 |
|  | $k_{z0}$ | 10 | 3.85 | 4.5 | 7.3 |
|  | $k_{rveg}$ | 10 | 1.36 | 7.01 | 8.04 |
|  | $k_{emis}$ | 10 | 2.39 | 3.62 | 6.47 |
|  | $k_{albedo}$ | 10 | 1.02 | 2.58 | 7.85 |

(b)

Table 7. Experiment 2 (different amplitudes of random noise in the observations), using Harvard
Forest and Kruger Park sites. We present the Mean values for 500 experiments:  (a) The first two
columns give the computed fluxes prior the assimilation. The last three columns present the RMSE
(prior minus estimated) for a given level of noise added to the observations (10%, 30%, 50%). (b)
The first two columns give the noise interval (in %) introduced for each control parameter with
respect to the initial value of 1. The last three columns present the mean  error in % on the control
parameters for different level of noise (10%, 30%, 50%) added to the observations (LST)

|  | | RMSE | | | |
|---|---|---|---|---|---|
|  | **Fluxes** | **Experiment 3 (PFT 1)** | | **Experiment 4 (PFT 12)** | |
|  |  | **Prior** | **Final** | **Prior** | **Final** |
| Kruger Park | $H(W/m^2)$ | 4.22 | $4.1.10^{-12}$ | 2.18 | $2.4.10^{-9}$ |
|  | $LE(W/m^2)$ | 4.51 | $2.6.10^{-4}$ | 6.86 | $3.2.10^{-5}$ |
|  | $LST\,(K)$ | 7.15 | $2.3.10^{-5}$ | 2.32 | $8.3.10^{-9}$ |
| Harvard Forest | $H(W/m^2)$ | 2.33 | $2.2.10^{-12}$ | 1.52 | $1.5.10^{-10}$ |
|  | $LE(W/m^2)$ | 2.45 | $7.3.10^{-4}$ | 8.34 | $2.4.10^{-6}$ |
|  | $LST\,(K)$ | 5.14 | $4.3.10^{-5}$ | 1.37 | $7.1.10^{-10}$ |

(a)

|  | | Mean error (%) | | | |
|---|---|---|---|---|---|
|  | **Control Parameters** | **Experiment 3 (PFT 1)** | | **Experiment 4 (PFT 12)** | |
|  |  | **Prior** | **Final** | **Prior** | **Final** |
| Kruger Park | $k_{cond}$ | 25.4 | $3.17.10^{-11}$ | 27.3 | $6.37.10^{-6}$ |
|  | $k_{capa}$ | 25.3 | $3.1.10^{-11}$ | 27.3 | $5.64.10^{-6}$ |
|  | $k_{z0}$ | 25.1 | $6.7.10^{-11}$ | 26.3 | $7.97.10^{-5}$ |
|  | $k_{rveg}$ | - | - | 28.1 | $2.76.10^{-6}$ |
|  | $k_{emis}$ | 25.8 | $3.01.10^{-11}$ | 27.5 | $6.08.10^{-5}$ |
|  | $k_{albedo}$ | 25.9 | $5.2.10^{-11}$ | - | - |
| Harvard Forest | $k_{cond}$ | 24.1 | $5.58.10^{-5}$ | 26.9 | $5.85.10^{-6}$ |
|  | $k_{capa}$ | 25.4 | $5.57.10^{-6}$ | 25.8 | $7.84.10^{-7}$ |
|  | $k_{z0}$ | 24.4 | $1.27.10^{-5}$ | 25.8 | $7.84.10^{-7}$ |
|  | $k_{rveg}$ | - | - | 22.1 | $8.31.10^{-6}$ |
|  | $k_{emis}$ | 25.5 | $5.71.10^{-4}$ | 24.2 | $5.96.10^{-7}$ |
|  | $k_{albedo}$ | 23.4 | $1.99.10^{-4}$ | - | - |

(b)

Table 8. Results for Experiments 3 (PFT 1) and 4 (PFT 12). RMSE of model fluxes (a) and Parameters Relative errors (b) before and after the assimilation process on FLUXNET Harvard Forest and Kruger Park.

| | | RMSE | |
| | | Experiment 5 (PFT 12) | |
| | Fluxes | Prior | Final |
|---|---|---|---|
| Kruger Park | $H(W/m^2)$ | 30.4 | 2.1 |
| | $LE(W/m^2)$ | 34.1 | 3.1 |
| | $LST\ (K)$ | 3.12 | $3.2.10^{-1}$ |
| Harvard Forest | $H(W/m^2)$ | 41.5 | 5.4 |
| | $LE(W/m^2)$ | 24.1 | 2.3 |
| | $LST\ (K)$ | 5.2 | $5.1.10^{-1}$ |

(a)

3
4

| | | Mean error (%) | |
| | Control Parameters | Experiment 5 (PFT 12) | |
| | | Prior | Final |
|---|---|---|---|
| Kruger Park | $k_{cond}$ | 23.4 | $2.3.10^{-1}$ |
| | $k_{capa}$ | 26.6 | $2.1.10^{-1}$ |
| | $k_{z0}$ | 22.2 | $1.5.10^{-1}$ |
| | $k_{rveg}$ | 25.9 | $3.1.10^{-1}$ |
| | $k_{emis}$ | 24.5 | $2.3.10^{-1}$ |
| | $k_{albedo}$ | 23.8 | $1.8.10^{-1}$ |
| | $mx_{eau}$ | 26.3 | $6.8.10^{-1}$ |
| | $hum_{cste}$ | 22.4 | $1.9.10^{-1}$ |
| | $dpu_{cste}$ | 25.6 | $3.2.10^{-1}$ |
| | $rsol_{cste}$ | 23.1 | $1.9.10^{-1}$ |
| Harvard Forest | $k_{cond}$ | 25.1 | $3.30.10^{-1}$ |
| | $k_{capa}$ | 26.7 | $2.61.10^{-1}$ |
| | $k_{z0}$ | 25.4 | $1.79.10^{-1}$ |
| | $k_{rveg}$ | 27.5 | $2.8.10^{-1}$ |
| | $k_{emis}$ | 26.3 | $2.1.10^{-1}$ |
| | $k_{albedo}$ | 24.7 | $2.37.10^{-1}$ |
| | $mx_{eau}$ | 25.8 | $7.34.10^{-1}$ |
| | $hum_{cste}$ | 25.2 | $2.7.10^{-1}$ |
| | $dpu_{cste}$ | 24.2 | $2.2.10^{-1}$ |
| | $rsol_{cste}$ | 25.4 | $2.36.10^{-1}$ |

(b)

Table 9. Results for Experiment 5 (PFT 12). RMSE of model fluxes (a) and Parameters Relative
errors (b) before and after the assimilation process, on FLUXNET Harvard Forest and Kruger Park
sites.

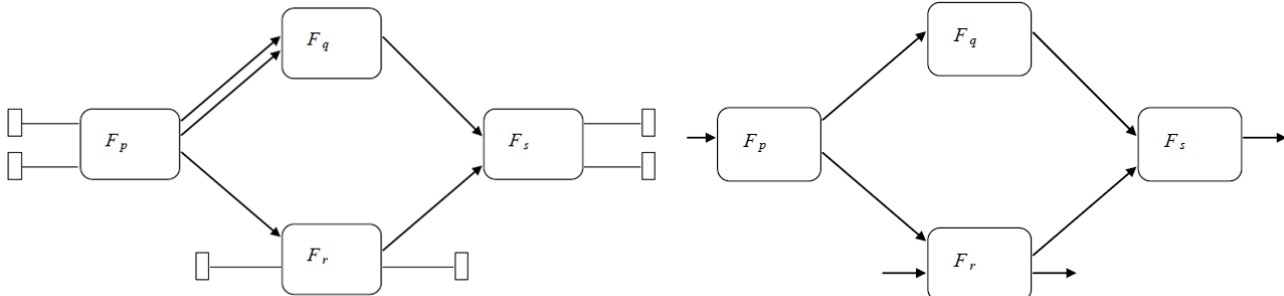

2     Figure 1 (left) Example of a modular graph associated with four basic functions and five basic

3     connections, three inputs points and three output points; (right) simplified description showing the

4     acyclicity of the graph. Source: Nardi et al, 2009

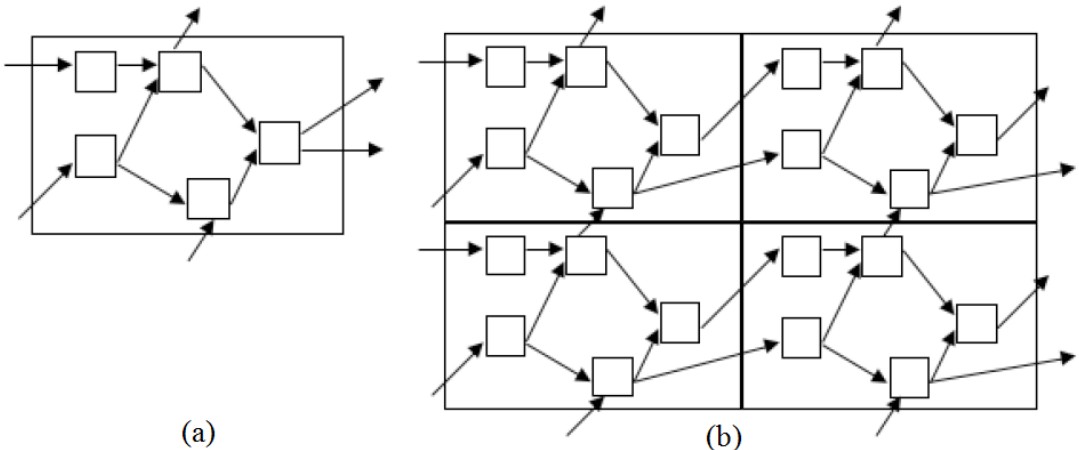

(a)  (b)

2  Figure 2. (a) Example of a modular graph with five modules, assumed representative of the

3  pointwise equations of a given model; (b) Partial view of the replication of the graph in space. Each

4  elementary graph with five modules is associated with one grid point. Source: Nardi et al, 2009

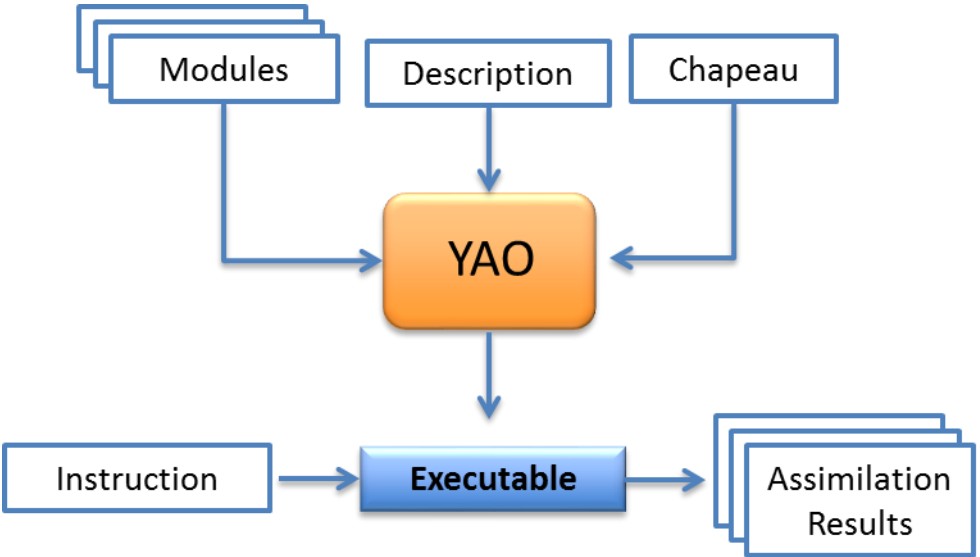

Figure 3. Structure of a project in YAO. The software generates an executable program from input modules, hat and description files. The generated program reads an instruction file to perform assimilation experiments.

REMPLACER CHAPEAU PAR HAT

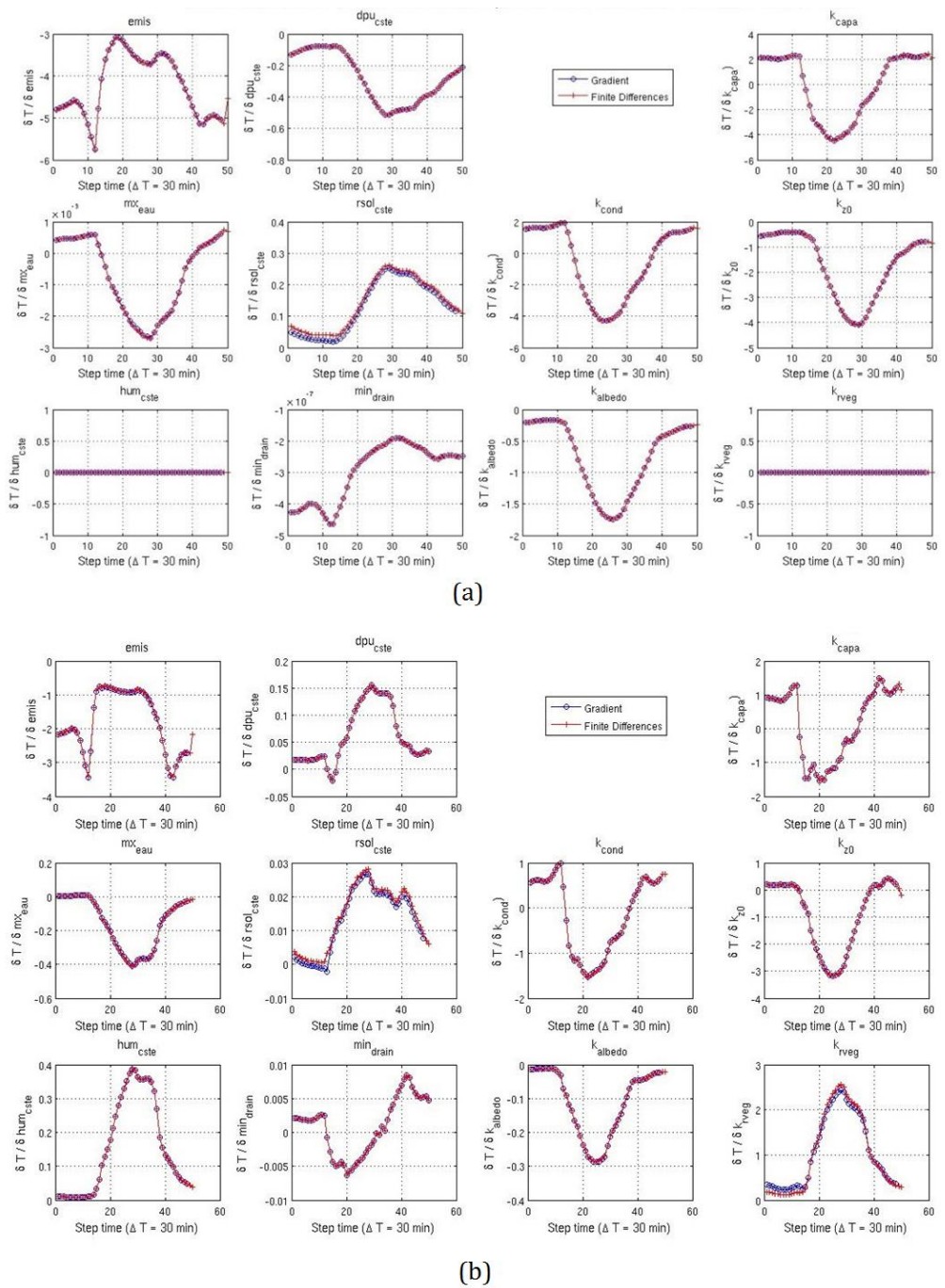

(a)

(b)

Figure 4. Comparisons for August 28,1996 at Harvard Forest, of the sensitivities obtained for each
control parameter with both the finite differences and the model gradients computed with the adjoint
model. Sensitivity analysis results for PFT 1 are in Fig.4 (a) and  for PFT 12 in Fig.4(b). The
sensitivities were computed on the surface temperature for Harvard Forest.  Blue curves represent the
LST derivative with respect to each parameter given by the adjoint each half hour over a day. Red
curves represent the LST derivative computed with a finite difference discretization of the model.

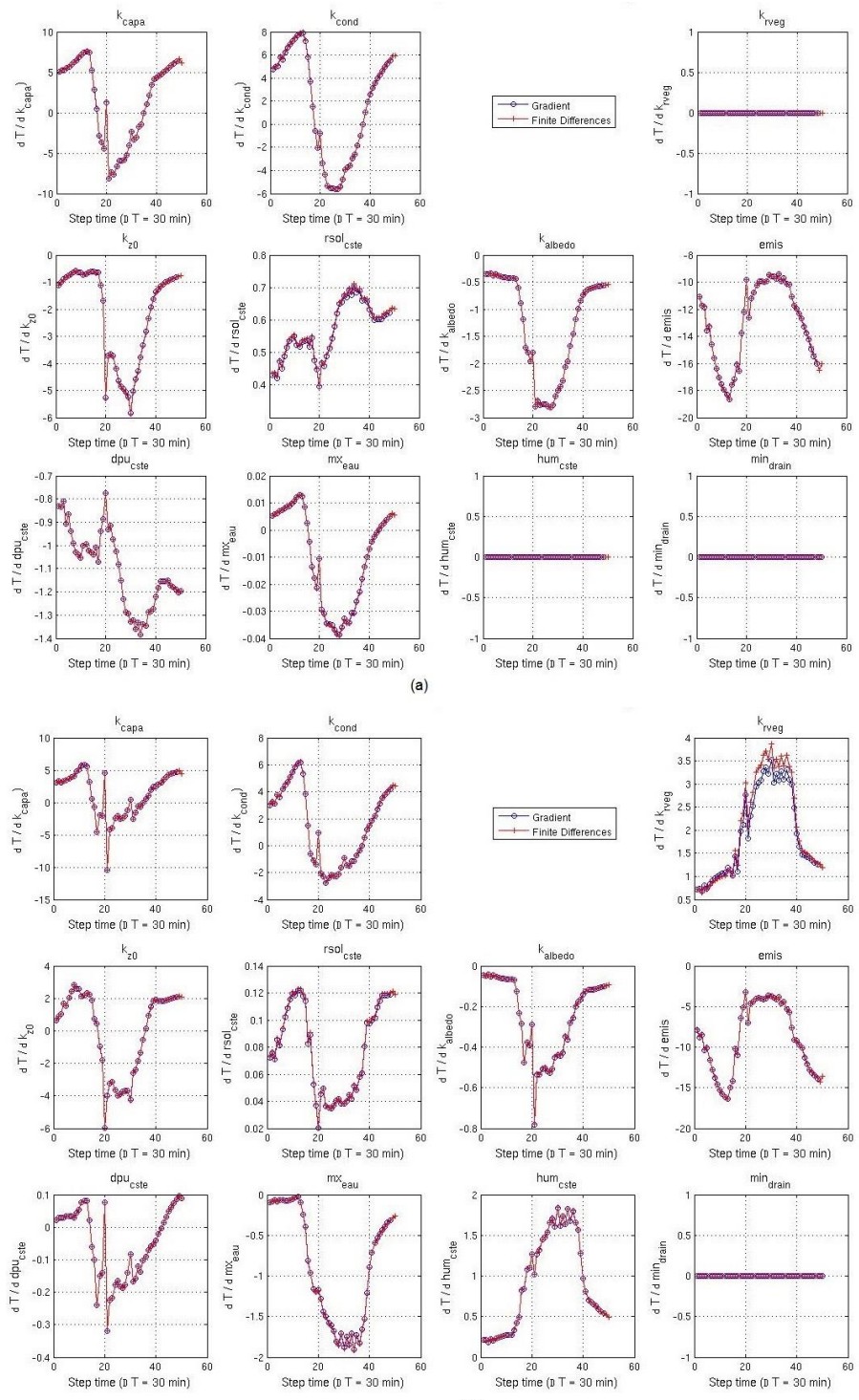

Figure 5. Comparisons for February 11,2003 Kruger Park site, of the sensitivities obtained for each control parameter with both the finite differences and the model gradients computed with the adjoint model. Sensitivity analysis results for PFT 1 are in Fig.5 (a) and for PFT 12 in Fig.5(b). The sensitivities were computed on the surface temperature. Blue curves represent the LST derivative with respect to each parameter given by the adjoint each half hour over a day. Red curves represent the LST derivative computed with a finite difference discretization of the model.

APPENDIX A

**SECHIBA-YAO**

The version of SECHIBA implemented in YAO includes the two-layer hydrology of Choisnel (1977), mentioned in Section 2. SECHIBA original code is implemented in a modular scheme, having a set of well-defined routines, independent in its processes and with a single entry point (a main routines handling the rest of the functionalities).

 A set of prognostic variables is defined for each module and its assignation depends on the forcing conditions, physics phenomena, etc. SECHIBA can work coupled with the other components of ORCHIDEE (STOMATE and LPJ) or it can be used offline, as it was used in this work. Once SECHIBA is coded in YAO, it can be easily coupled with the other modules of ORCHIDEE.

In SECHIBA, the different routines were originally coded in the Fortran language and can be run at any resolution and over any region of the globe. The version of SECHIBA implemented in YAO is denoted SECHIBA-YAO and follows the Fortran code. In its present form, it can only be run at one point at a time.

ORCHIDEE uses MODIPSL and IOIPSL in its internal processes (see http://forge.ipsl.jussieu.fr/igcmg/wiki/platform/documentation for more information). Developed at IPSL, the first one is a set of scripts allowing the extraction of a given configuration from a computing machine and the compilation of the specific machine configuration components. MODIPSL is the tree that will host models and tools for configuration. IOIPSL helps to manage variables state history, variable normalization, file lecture, and among others.

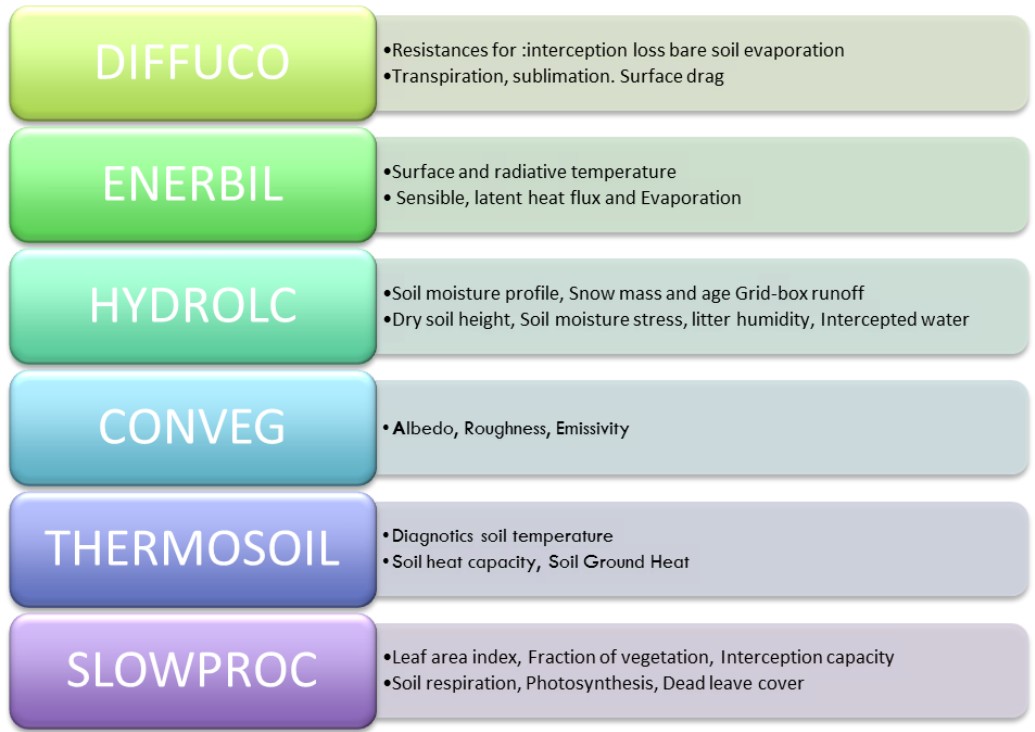

Figure A1 SECHIBA subroutines and its corresponding outputs. Source: Benavides, 2014.

The main routines in SECHIBA-Fortran are presented in Fig A1. These are also the routines considered in the YAO implementation of the model. First, DIFFUCO computes the diffusion and plant transpiration coefficients based on the atmospheric conditions, solar fluxes, dry soil height, soil moisture stress and fraction of vegetation. ENERBIL corresponds to the energy budget module. Surface energy fluxes related to the soil are computed, based on atmospheric conditions, radiative fluxes, resistances, surface type fractions and surface drag. HYDROLC is the hydrological budget module, taking as inputs the rainfall, snowfall, evaporation components, soil temperature profile and vegetation distribution. CONDVEG helps in the computation of the vegetation conditions. The thermodynamics of the model is computed in THERMOSOIL, based on a seven-layer soil profile. Finally, SLOWPROC computes the soil slow processes. When SECHIBA is decoupled from STOMATE, this module deals also with the LAI evolution.

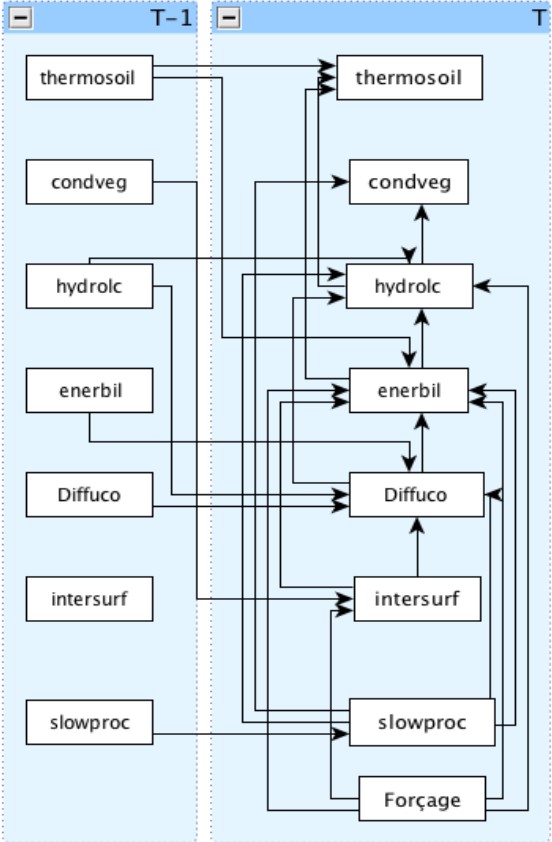

Figure A2 SECHIBA hyper graph, showing general model dynamics. Source: Benavides, 2014

The different SECHIBA components are interconnected as shown in Fig.A2. The output of the different modules serves as inputs for the next one, thus resulting in an interdependency among modules to be considered when modeling SECHIBA-YAO.