# Peer review of "Variational Assimilation of Land Surface Temperature within the ORCHIDEE Land Surface Model Version 1.2.6"

_Geoscientific Model Development, 2016_

## Short Comment (SC1) · 15 Apr 2016

Dear Authors,

as Executive Editor of GMD I appreciate that you comply with the Editorial of GMD by including the model name and version number in the title of your article. Nevertheless, reading the abstract and the Code Availability Section of your article shows, that the development published here is SECHIBA(-YAO). You already took the effort to name it, it would be good to specify a version label (an individual version number or other code version identifier) to this specific model part. As your publication focusses on SECHIBA it should be named and labeled in the title as well.

I suggest to change the title of your article in a way like

"Variational Assimilation of Land Surface Temperature within the ORCHIDEE Land Surface Model Version 1.2.6 using SECHIBA (vX.Y)"

upon revision.

Best regards,

Astrid Kerkweg

––––––––––––––––––––––––––

---

## Short Comment (SC2) · 15 Apr 2016

Dear Astrid,

thank you for your kindly comment. Actually SECHIBA version used in the work corresponds to the same version of ORCHIDEE, since ORCHIDEE is compose of several models (including SECHIBA of course) so all models have the same version number. However for SECHIBA-YAO, I agree with you that a model version has to be mentioned in the work. I will take this into account in the article

Regards!

Hector Benavides

---

## Referee Comment (RC1) · A. KALLEL (Referee) · 8 May 2016

**A. KALLEL (Referee)**

abdelaziz\_kallel@yahoo.fr

Received and published: 8 May 2016

Summary — The paper present a variational assimilation of the LST within the SECHIBA module of ORCHIDEE. The YAO software was used to compute the adjoint model than to implement the 4D-VAR assimilation. Experiments show the accuracy of the model. However, the authors do not explain how they simulate a priori and noise on LST data.

General comments ———— You said in section 3.1 that you use the Gradient algorithm but you do not explain what kind of algorithm it is exactly: is it "Levenberg-Marquardt algorithm" ?

You do not explain well how to estimate the actual control parameter values given the a priori. Indeed, the relationship prior value/actual value determines the covariance matrix B in Eq. (4)

In your experiments you do not add noise to observation so in this case R is 0 and Eq. (4) is not well defined (division by 0). For that I suggest to add noise to observation an study the robustness of the developed approach as a function of the noise level.

- Page 2: It is well known that both approaches pro-Detailed comments vide the same solution at the end of the assimilation period, for perfect and linear models. - -> It is well known that both approaches provide the same solution at the end of the assimilation period, for Gaussian variables, and perfect and linear models. Page 5, Line 23: index i is forgotten in epsilon Paga 8, line 32: you said "the second approach was used". I do not understand what is it "the second approach". Page 9, line 11: you said "the initial model". Same problem, I do not understand. Page 9, line 25: "the parameter prior values were retrieved successfully." In general, we estimate the actual values and not the prior. The prior is what we know initially before observation. Page 12, line 24: more difficult it is to find local minima that correspond to the initial control parameters values - -> more difficult it is to find global minima that correspond to the initial control parameters values Page 12, line 25: It is difficult to retrieved parameters - -> It is difficult to retrieve parameters Page 12, line 26: the assimilation of this variable in order to optimize these parameters is not optimal - -> the assimilation of this variable in order to optimize these parameters is not efficient

---

## Referee Comment (RC2) · R. Mechri (Referee) · 9 May 2016

Dear Authors,

I have appreciated the global work and I find the topic really interesting especially the use of the variational method in a double goal : the sensitivity analysis and the parameter estimation. I think that some minor corrections should be performed before being published and here are my corrections and my comments on this work.

[Figure]

**1 Minor corrections**

- Abstract

  1. The sentence corresponding to page 1, lines 16 to 18 is too long and should be shortened or divided in two sentences.

- Section 2 : Models and Data

  1. Page 3, line 16 change "$22^{th}$" to "$22^{nd}$".
  2. Page 4, line 10 the unit is not clear for the spectral band "$\mu m$".

- Section 3 : The Methodology

  – Subsection 3.1: Variational Assimilation

    1. Page 6, line 3 : you should replace the "$f$" at the end of equation (5) by "$J$"
    2. In the page 6, lines 6 and 7 you explained that y is described by equation 2. I can't see the relationship between equation 2 describing the empirical formulation of the brightness temperature and the surface radiation and the description of the observation term "$y$". Are you making reference to the equation described in page 5 at line 23 ?

  – Subsection 3.4: Development of SECHIBA-YAO

    1. Page 8, line 11 : change "ANNEX A" to "Appendix A"

- Data assimilation experiments

  – Subsection 4.3: Experiment Definition

    1. Page 11, line 27 : change as follows : "sensible (H) and latent (LE) heat".

– Subsection 4.4: Experiment Definition

 1. Page 12, line 25 : correct 'retrieved' to 'retrieve'.

**2   Questions and Comments**

- General comments

 1. I think that you should write the corresponding DOI and pages numbers for the different references used.

 2. Figures are not clear when printed in A4 format and legends are almost not visible I think you should use larger size for the legends and make your figures in a vectorial (.ps, .eps, .pdf, etc.) format so that the zoom could be possible without degrading the figures quality.

- Specific comments

 1. In the variational assimilation can you please specify what do you exactly mean by observations and first guess : what are you exactly assimilating **Pnoise** ( referred as 'first guess' and 'perturbed' in figures 5 and 6 (a and b) ) or **Ptrue** (referred as 'observations' and 'initial value' in figures 5 and 6 (a and b)) ?

   (a) In the case you are assimilation observations then how could you perform your validation using the same observations?

   (b) In the case you are assimilation your **Pnoise** then can you explain more how did you perturbed the 'Truth' using your uniform random noise (precise the respective variation ranges of the different assimilated variables so that we can see how much $50\%$ of the nominal value is consistent ) ?

2. In the experiment 3 the gal was to show how could the number of variables included in the assimilation affects the performances of the method. In this case Experiment 3 must have the same conditions than Experiment 2 except the number of assimilated variables. Surprisingly you have changed the assimilation period starting the 8th of August 1996 rather than the $3^{rd}$ of March. My questions are the following :

   (a) Why did you change the starting date of the assimilation?

   (b) How could you know that the decrease in the performances is only related to the number of parameters knowing that you have taken a different assimilation period and knowing the fact that the sensitivity of parameters toward LST is - as you have already mentioned- dependent on the seasons, period of the day etc. ?

---

## Referee Comment (RC3) · B. Coudert (Referee) · 18 May 2016

**B. Coudert (Referee)**

benoit.coudert@cesbio.cnes.fr

Received and published: 18 May 2016

The article addresses a 4D-VAR assimilation experiment of synthetic LST with YAO software in the SECHIBA module of the ORCHIDEE Land Surface Model developed at the "Institut Pierre Simon Laplace (IPSL)". The authors focus on the presentation of SECHIBA-YAO model and methodology with the stake to deliver an assimilation toolbox for end users. Twin experiments are conducted with synthetic data (for assimilation) in order to determine the most sensitive parameters of the model through 3 experiments on bare soil and C3 crop. Their study aims to demonstrate i) the ability of the assimilation methodology and ii) the potential of the LST in the retrieval of the 10 parameters and initial conditions to control leading consequently to the improvement

of surface fluxes and LST estimates.

General Comments

The paper is clearly organized and quite easy to follow. I have appreciated the global approach. Nevertheless, some precisions and complements are expected in the results interpretation. The authors should also precise which variable is really used for the assimilation: brightness temperature in the [8-14] microns interval, surface radiance or LST ? Why one or the other is finally chosen ? (See specific comments). The good results of the methodology have to be nuanced by the choice of the dataset used to manage the study and also by the fact that synthetic data are used (see specific comments below). I encourage the authors to carry on with their study with the difficult but interesting transition to the use of real remote sensed data. It is also expected that the authors take care to place their work among the different studies dealing with the control of LSM or SVAT models with surface temperature for water budget or surface fluxes estimates purposes.

Some Specific Comments

Section 1: P.2, L.31: could you precise what is a "specific deep land surface temperature" ? P3, L.1: "or" should be replace with "of" P3, L3: remove "available"

Section 2: P4., L.34-36: The SECHIBA version used has a "two-layer soil profile" meanwhile in appendix A (P.28, L.8-9) a "seven-layer soil profile" is mentioned for the THERMOSOIL subroutine. Please bring some precisions or corrections.

P.4.: L.1-12: could you precise why do you prefer the use of a brightness temperature in the interval [8-14] microns instead of the LST ? I certainly misunderstand the explanation.

L.6, Eq.1: the Stefan Boltzmann constant [sigma] has been omitted in the first term of the equation. L.6, Eq.1: is LW\_down estimated or measured in situ ? In this case, could you precise the spectral band associated and if a band factor has been applied to

take into account that only a fraction of the radiation is measured in the spectral interval according to the Planck's law at the difference of the Stefan-Boltzmann law. Precisions are thus required regarding the use of the Svendsen conversion function (Eq.2).

Table 3, P.18: "LST" is mentioned as observation but is it: LST, radiance or brightness temperature in the [8-14] microns interval ? You should also indicate that it is a synthetic observation.

P.4, L24: could you precise what is the type of the C3 crop for both sites and also give some details on the phenology or state of the plant development. As an example, LAI and canopy height could be added in Table 3 for PFT12.

Section 3: P.6, L.3, Eq.5: the cost function "f" should be replace with "J" in relation to Eq.4 P.6, L.7: I suppose that "y" should be replaced with "J". I do not understand the reference to equation 2 which is the expression of the brightness temperature...

P.7, L.32: this empty line should be suppressed.

P.8, L18-19: the sentence is unclear, please correct the syntax.

P.8, L.32: "the second approach was used"... I certainly miss something but you have not presented several approaches in this subsection.

Section 4: P.9, L.16: "The other parameters are multiplicative factors". Why don't you consider directly the parameters themselves: surface emissivity instead of kemis, albedo instead of kalbedo, ... etc. ? Is it only due to a technical (or numerical) reason ?

P.9, L.23: instead of "optimal value", you certainly mean "initial value" ?

P.10, L.5-6 and Table 1 (P.16): the initial value of mxeau (maximum water content) parameter is very low (150kg/m3). Why this choice ? What types of soil are considered ? It is important to mention somewhere the soil description (classification or texture). A low mxeau value corresponds to dry or stressed surface conditions and will consequently increase the LST and overestimate it compared to in situ measurements. This

СЗ

remark is confirmed by the LE times series of figures 5&6 (see comments below) with quasi null absolute values. Is it done to increase the parameter sensitivity to LST in order to improve the results ?

P.10, L.26-29: in order to facilitate the interpretation of the results of Figure 4 and Table 2, you should precise earlier how the parameter sensitivity hierarchy is defined with both methodologies (finite differences and model gradients), i.e. based on the slope of the gradients.

P.11, L.12-18: you should homogenize your notations throughout the text, tables and figures ("true" = observation, "noise" = first guest or perturbed, "assim"= after assimilation) in order to clarify.

P.12, section 4.4 "Results" and Tables 4 and 5: could you explain how a RMSD on LST reaching 5K is compatible with RMSD on surface fluxes lower than 2.5 W/m2 for experiment 1? The same could be addressed for experiment 2 although RMSD on LST is lower and RMSD on LE higher (but even though relatively low in absolute value).

Figures 5&6: times series of LE for bare soil and although for C3 crop have very low absolute values (less than 5W/m2). It is related to the low mxeau value (see previous comment)? Are the synthetic observations times series realistic compared to real observations? You should give more information on these points in order to argue your choices and to comment the physical behavior of the model. From a physical point of view, I am surprised by the fact that times series are similar for figures 5 (bare soil) and 6 (C3 crop). During the simulation period of 7 days, LST increases by about 10K meanwhile H flux decrease and LE flux stays quasi null...how is it possible? Times series of meteorological forcing and a description of the vegetation development should be helpful for the analysis.

Section 5: P.13, L.1: "LST" should be replaced with "synthetic LST".

Conclusion:

A correctly revised manuscript answering the questions and bringing the required precisions and corrections might certainly be of interest to the readership of GMD.

---

## Short Comment (SC3) · 30 May 2016

Dear Rihab,

Thank you for your review and for the interest in our work. I make list of answers regarding all your comments and questions Minor corrections • Abstract 1. The sentence corresponding to page 1, lines 16 to 18 is too long and should be shortened or divided in two sentences. Remark taken into account. The phrase will be replaced from the manuscript to the following sentence: SECHIBA-YAO allows the control of the eleven most influent internal parameters or initial conditions of the soil water content, by observing the land surface temperature or remote sensing data as brightness temperature.

• Section 2 : Models and Data 1. Page 3, line 16 change "22th" to "22nd". Modification taken into account 2. Page 4, line 10 the unit is not clear for the spectral band "um". It is the spectral band $\mu$m. Modification taken into account

• Section 3 : The Methodology –Subsection 3.1: Variational Assimilation 1. Page 6, line 3 : you should replace the "f" at the end of equation (5) by "J" Modification taken into account 2. In the page 6, lines 6 and 7 you explained that y is described by equation 2. I can't see the relationship between equation 2 describing the empirical formulation of the brightness temperature and the surface radiation and the description of the observation term "y". Are you making reference to the equation described in page 5 at line 23 ? Eq (2) makes reference to the calculation of brithness temperature in SECHIBA based on the Radiation, term that can be later used as observation if needed. The reference to equation 2 is misplaced, it will be erased from the manuscript –Subsection 3.4: Development of SECHIBA-YAO 1. Page 8, line 11 : change "ANNEX A" to "Appendix A" Modification taken into account

• Data assimilation experiments –Subsection 4.3: Experiment Definition 1. Page 11, line 27 : change as follows : "sensible (H) and latent (LE) heat Modification taken into account –Subsection 4.4: Experiment Definition 1. Page 12, line 25 : correct 'retrieved' to 'retrieve'. Modification taken into account Questions and Comments Regarding the questions, I make a point by point answer to all your different comments.

1. In the variational assimilation can you please specify what do you exactly mean by observations and first guess : what are you exactly assimilating Pnoise ( referred as 'first guess' and 'perturbed' in figures 5 and 6 (a and b)) or Ptrue (referred as 'observations' and 'initial value' in figures 5 and 6 (a and b)) ? Since we are performing twin experiments, an initial set of parameters (Ptrue) is used to produce synthetic observations. The idea is to perturbate Ptrue (to obtained Pnoise,meaning my first guess). The idea is to used the synthetic observations produced with Ptrue in order to go from Pnoise to Ptrue by the asismilation process. (a) In the case you are assimilating observations then how could you perform your validation using the same observations?

Since the assimilation process estimates control parameters, its final values will affect the final model state, thus a comparison between observations and the final estimated temperature allows to validate the assimilation (b) In the case you are assimilation your Pnoise then can you explain more how did you perturbed the 'Truth' using your uniform random noise (precise the respective variation ranges of the different assimilated variables so that we can see how much 50% of the nominal value is consistent ) ? The perturbation was a random noise produce by the machine, limited up to 50% the true parameter value, equal to one, so the perturbed value is constrained between [0.5 , 1.5]

2. In the experiment 3 the goal was to show how could the number of variables included in the assimilation affects the performances of the method. In this case Experiment 3 must have the same conditions than Experiment 2 except the number of assimilated variables. Surprisingly you have changed the assimilation period starting the 8th of August 1996 rather than the 3rd of March. My questions are the following : (a) Why did you change the starting date of the assimilation? I want to test different conditions in the assimilation capabilities. The scope of this work was only to prove the assimilation potential. More experiments have been done and give similar results. For further informations and tests, readers can consult my thesis (Benavides, 2014). (b) How could you know that the decrease in the performances is only related to the number of parameters knowing that you have taken a different assimilation period and knowing the fact that the sensitivity of parameters toward LST is - as you have already mentioned-dependent on the seasons, period of the day etc. ? The decrease in performance is related to the complexity of the cost function used during the assimilation process: the more the number of parameters the more complex the cost function is. A decrease during the assimilation can be expected, regardless the season, period of the day, etc. In my thesis (Benavides, 2014) I performed other experiments that corroborates this statement. I have added a sentence in the manuscript explaining the general validity of the result

---

## Author Response (AR1)

**1. REPONSE TO BENOIT COUDERT**

Dear Reviewer,

Thank you for your review and for the interest in our work. I make list of answers regarding all your comments and questions

**SPECIFIC COMMENTS**

**Section 1:**

**P.2, L.31: could you precise what is a "specific deep land surface temperature" ?**

The sentence has been changed in the revised version because it was wrong and related to another paper not referenced here. The new sentence is now: "When assimilating LST into the model, the authors proved that the assimilation of LST can improve the model simulated heat and water fluxes. "

**P3, L.1: "or" should be replace with "of"**

Modification taken into account

**P3, L3: remove "available"**

Modification taken into account

**Section 2:**

**P4., L.34-36: The SECHIBA version used has a "two-layer soil profile" meanwhile in appendix A (P.28, L.8-9) a "seven-layer soil profile" is mentioned for the THERMOSOIL subroutine. Please bring some precisions or corrections.**

A two-layer hydrology was used in this ORCHIDEE version. The seven layer discretization is for the resolution of the heat diffusion equation. We have changed the text in the paper to make it clearer

**P.4.: L.1-12: could you precise why do you prefer the use of a brightness temperature in the interval [8-14] microns instead of the LST ? I certainly misunderstand the explanation.**

The use of this variable follows my previous thesis work (Benavides, 2014) when observations coming from a thermal infrared radiometer were used as observations (SMOSREX). This interval correspond to the radiometer filter used for these measurements.

L.6, Eq.1: the Stefan Boltzmann constant [sigma] has been omitted in the first term of the equation. L.6, Eq.1: is LW\_down estimated or measured in situ ? In this case, could you precise the spectral band associated and if a band factor has been applied to take into account that only a fraction of the radiation is measured in the spectral interval according to the Planck's law at the difference of the Stefan-Boltzmann law. Precisions are thus required regarding the use of the Svendsen conversion function (Eq.2).

We don't understand your remark: in equation 1, we wrote the total radiation emitted by a soil surface and integrated on all the long wave spectra. The SB constant don't appear on the left side of the equation. In our case, LW downward is measured by a large band radiometer and this is why we can use the Svendsen's formula to estimate LST. The manuscript has been revised to clarify the notations and the confusions between LST and TB.

**Table 3, P.18: "LST" is mentioned as observation but is it: LST, radiance or brightness temperature in the [8-14] microns interval ? You should also indicate that it is a synthetic observation.**

I can assimilate LST or TB computed from a radiometer measurements. In my distributed version only LST observations are included. In the full SECHIBA-YAO version both measurements can be chosen.

**P.4, L24: could you precise what is the type of the C3 crop for both sites and also give some details on the phenology or state of the plant development. As an example, LAI and canopy height could be added in Table 3 for PFT12.**

Vegetation in ORCHIDEE is characterized by using Plant Functional Type system of classification. Although PFT system describes to types of cultures (C3 and C4crop) it does not distinguish varieties of crops and only one crop type is currently active

**Section 3:**

**P.6, L.3, Eq.5: the cost function "f" should be replace with "J" in relation to Eq.4**

Modification taken into account

**P.6, L.7: I suppose that "y" should be replaced with "J". I do not understand the reference to equation 2 which is the expression of the brightness temperature**

Reference to equation 2 misplaced. Modification taken into account

**P.7, L.32: this empty line should be suppressed.**

Modification taken into account

**P.8, L18-19: the sentence is unclear, please correct the syntax.**

The phrase will be replaced by: "When studying the subroutines, their complexity was reduced by breaking the different steps into simpler elements."

**P.8, L.32: "the second approach was used" I certainly miss something but you have not presented several approaches in this subsection.**

Misplaced reference: this sentence will be erased

**Section 4: P.9, L.16: "The other parameters are multiplicative factors". Why don't you consider directly the parameters themselves: surface emissivity instead of kemis, albedo instead of kalbedo, etc. ? Is it only due to a technical (or numerical) reason ?**

The idea is to have all parameters with the same value (all equal to 1), in order to have directly the magnitude of the assimilation quality, and with the idea of having the possibility of comparing them

**P.9, L.23: instead of "optimal value", you certainly mean "initial value" ?**

What I meant is that prior to assimilation and to any perturbation, model parameters are always equal to 1

**P.10, L.5-6 and Table 1 (P.16): the initial value of mxeau (maximum water content) parameter is very low (150kg/m3). Why this choice ?**

This is the initial value generally used in sechiba before spinup.

What types of soil are considered? It is important to mention somewhere the soil description (classification or texture).

Yes, you are true, the soil texture has been added in the text.

A low mxeau value corresponds to dry or stressed surface conditions and will consequently increase the LST and overestimate it compared to in situ measurements. This remark is confirmed by the LE times series of figures 5&6 (see comments below) with quasi null absolute values. Is it done to increase the parameter sensitivity to LST in order to improve the results ?

Yes, we agree, and this is the case in our experiments, we took dry conditions to be close to the initial value prescribed in Sechiba, but we could have chosen another value. This is at this stage only synthetic observations and twin experiments. The next step is the assimilation of actual observations which will be our future work.

**P.10, L.26-29: in order to facilitate the interpretation of the results of Figure 4 and Table 2, you should precise earlier how the parameter sensitivity hierarchy is defined with both methodologies (finite differences and model gradients), i.e. based on the slope of the gradients.**

I didn't want to give much details on this because I think is out of the scope of the work: However I give a reference to my thesis (Benavides, 2014), where I give much details regarding this remark. However I clarified this point in the final manuscript

P.11, L.12-18: you should homogenize your notations throughout the text, tables and figures ("true" = observation, "noise" = first guest or perturbed, "assim"= after assimilation) in order to clarify.

Modification taken into account

P.12, section 4.4 "Results" and Tables 4 and 5: could you explain how a RMSD on LST reaching 5K is compatible with RMSD on surface fluxes lower than 2.5 W/m2 for experiment 1? The same could be addressed for experiment 2 although RMSD on LST is lower and RMSD on LE higher (but even though

relatively low in absolute value). Figures 5&6: times series of LE for bare soil and although for C3 crop have very low absolute values (less than 5W/m2). It is related to the low mxeau value (see previous comment) ? Are the synthetic observations times series realistic compared to real observations ? You should give more information on these points in order to argue your choices and to comment the physical behavior of the model. From a physical point of view, I am surprised by the fact that times series are similar for figures 5 (bare soil) and 6 (C3 crop). During the simulation period of 7 days, LST increases by about 10K meanwhile H flux decrease and LE flux stays quasi null how is it possible ? Times series of meteorological forcing and a description of the vegetation development should be helpful for the analysis.

The experiments have been done in dry soil conditions, close to the initial value prescribed in Sechiba. We remind that we present here twin experiments, to present the tools developed and their potentialities. The dry soil conditions explain why there is not much difference between bare soil and C3crop with very low evapotranspiration rates. During this period, the ground heat flux increases and heat the soil, explaining the increase of the Surface temperature.

Section 5: P.13, L.1: "LST" should be replaced with "synthetic LST".

done

**2. REPONSE TO RIHAB MECHRI**

Dear Reviewer,

Thank you for your review and for the interest in our work. I make list of answers regarding all your comments and questions

**SPECIFIC COMMENTS**

**Section 1:**

**P.2, L.31: could you precise what is a "specific deep land surface temperature" ?**

The sentence has been changed in the revised version because it was wrong and related to another paper not referenced here. The new sentence is now: "When assimilating LST into the model, the authors proved that the assimilation of LST can improve the model simulated heat and water fluxes. "

**P3, L.1: "or" should be replace with "of"**

Modification taken into account

**P3, L3: remove "available"**

Modification taken into account

**Section 2:**

P4., L.34-36: The SECHIBA version used has a "two-layer soil profile" meanwhile in appendix A (P.28, L.8-9) a "seven-layer soil profile" is mentioned for the THERMOSOIL subroutine. Please bring some precisions or corrections.

A two-layer hydrology was used in this ORCHIDEE version. The seven layer discretization is for the resolution of the heat diffusion equation. We have changed the text in the paper to make it clearer

**P.4.: L.1-12: could you precise why do you prefer the use of a brightness temperature in the interval [8-14] microns instead of the LST ? I certainly misunderstand the explanation.**

The use of this variable follows my previous thesis work (Benavides, 2014) when observations coming from a thermal infrared radiometer were used as observations (SMOSREX). This interval correspond to the radiometer filter used for these measurements.

L.6, Eq.1: the Stefan Boltzmann constant [sigma] has been omitted in the first term of the equation. L.6, Eq.1: is LW\_down estimated or measured in situ ? In this case, could you precise the spectral band associated and if a band factor has been applied to take into account that only a fraction of the radiation is measured in the spectral interval according to the Planck's law at the difference of the Stefan-Boltzmann law. Precisions are thus required regarding the use of the Svendsen conversion function (Eq.2).

We don't understand your remark: in equation 1, we wrote the total radiation emitted by a soil surface and integrated on all the long wave spectra. The SB constant don't appear on the left side of the equation. In our case, LW downward is measured by a large band radiometer and this is why we can use the Svendsen's formula to estimate LST. The manuscript has been revised to clarify the notations and the confusions between LST and TB.

**Table 3, P.18: "LST" is mentioned as observation but is it: LST, radiance or brightness temperature in the [8-14] microns interval ? You should also indicate that it is a synthetic observation.**

I can assimilate LST or TB computed from a radiometer measurements. In my distributed version only LST observations are included. In the full SECHIBA-YAO version both measurements can be chosen.

**P.4, L24: could you precise what is the type of the C3 crop for both sites and also give some details on the phenology or state of the plant development. As an example, LAI and canopy height could be added in Table 3 for PFT12.**

Vegetation in ORCHIDEE is characterized by using Plant Functional Type system of classification. Although PFT system describes to types of cultures (C3 and C4crop) it does not distinguish varieties of crops and only one crop type is currently active

**Section 3:**

**P.6, L.3, Eq.5: the cost function "f" should be replace with "J" in relation to Eq.4**

Modification taken into account

**P.6, L.7: I suppose that "y" should be replaced with "J". I do not understand the reference to equation 2 which is the expression of the brightness temperature**

Reference to equation 2 misplaced. Modification taken into account

**P.7, L.32: this empty line should be suppressed.**

Modification taken into account

**P.8, L18-19: the sentence is unclear, please correct the syntax.**

The phrase will be replaced by: "When studying the subroutines, their complexity was reduced by breaking the different steps into simpler elements."

**P.8, L.32: "the second approach was used" I certainly miss something but you have not presented several approaches in this subsection.**

Misplaced reference: this sentence will be erased

**Section 4: P.9, L.16: "The other parameters are multiplicative factors". Why don't you consider directly the parameters themselves: surface emissivity instead of kemis, albedo instead of kalbedo, etc. ? Is it only due to a technical (or numerical) reason ?**

The idea is to have all parameters with the same value (all equal to 1), in order to have directly the magnitude of the assimilation quality, and with the idea of having the possibility of comparing them

**P.9, L.23: instead of "optimal value", you certainly mean "initial value" ?**

What I meant is that prior to assimilation and to any perturbation, model parameters are always equal to 1

**P.10, L.5-6 and Table 1 (P.16): the initial value of mxeau (maximum water content) parameter is very low (150kg/m3). Why this choice ?**

This is the initial value generally used in sechiba before spinup.

What types of soil are considered? It is important to mention somewhere the soil description (classification or texture).

Yes, you are true, the soil texture has been added in the text.

A low mxeau value corresponds to dry or stressed surface conditions and will consequently increase the LST and overestimate it compared to in situ measurements. This remark is confirmed by the LE times series of figures 5&6 (see comments below) with quasi null absolute values. Is it done to increase the parameter sensitivity to LST in order to improve the results ?

Yes, we agree, and this is the case in our experiments, we took dry conditions to be close to the initial value prescribed in Sechiba, but we could have chosen another value. This is at this stage only synthetic observations and twin experiments. The next step is the assimilation of actual observations which will be our future work.

**P.10, L.26-29: in order to facilitate the interpretation of the results of Figure 4 and Table 2, you should precise earlier how the parameter sensitivity hierarchy is defined with both methodologies (finite differences and model gradients), i.e. based on the slope of the gradients.**

I didn't want to give much details on this because I think is out of the scope of the work: However I give a reference to my thesis (Benavides, 2014), where I give much details regarding this remark. However I clarified this point in the final manuscript

P.11, L.12-18: you should homogenize your notations throughout the text, tables and figures ("true" = observation, "noise" = first guest or perturbed, "assim"= after assimilation) in order to clarify.

Modification taken into account

P.12, section 4.4 "Results" and Tables 4 and 5: could you explain how a RMSD on LST reaching 5K is compatible with RMSD on surface fluxes lower than 2.5 W/m2 for experiment 1? The same could be addressed for experiment 2 although RMSD on LST is lower and RMSD on LE higher (but even though

relatively low in absolute value). Figures 5&6: times series of LE for bare soil and although for C3 crop have very low absolute values (less than 5W/m2). It is related to the low mxeau value (see previous comment) ? Are the synthetic observations times series realistic compared to real observations ? You should give more information on these points in order to argue your choices and to comment the physical behavior of the model. From a physical point of view, I am surprised by the fact that times series are similar for figures 5 (bare soil) and 6 (C3 crop). During the simulation period of 7 days, LST increases by about 10K meanwhile H flux decrease and LE flux stays quasi null how is it possible ? Times series of meteorological forcing and a description of the vegetation development should be helpful for the analysis.

The experiments have been done in dry soil conditions, close to the initial value prescribed in Sechiba. We remind that we present here twin experiments, to present the tools developed and their potentialities. The dry soil conditions explain why there is not much difference between bare soil and C3crop with very low evapotranspiration rates. During this period, the ground heat flux increases and heat the soil, explaining the increase of the Surface temperature.

Section 5: P.13, L.1: "LST" should be replaced with "synthetic LST".

done

**3. REPONSE TO ABDELAZIZ KALLEL**

Dear Abdelaziz,

Thank you for your review and for the interest in our work. I make list of answers regarding all your comments and questions

**You said in section 3.1 that you use the Gradient algorithm but you do not explain what kind of algorithm it is exactly: is it "Levenberg-Marquardt algorithm" ?**

Regarding the gradient algorithm, a minimiser called M1QN3 is used within YAO. It use q quasi-Newton technique (the L-BFGS method of J. Nocedal) with a dynamically updated scalar or diagonal preconditioner.

You do not explain well how to estimate the actual control parameter values given the a priori. Indeed, the relationship prior value/actual value determines the covariance matrix B in Eq. (4)

The Eq (4) is the most general form of the variational assimilation. I only give an introduction to this formula, but the estimation for the actual control parameter values are out of the scope of this work.

In your experiments you do not add noise to observation so in this case R is 0 and Eq.(4) is not well defined (division by 0). For that I suggest to add noise to observation an study the robustness of the developed approach as a function of the noise level.

The equation 4 is just the general form. In YAO R is by default the identity matrix so users can modify its value when necessary

**MODIFICATIONS TO THE MANUSCRIPT**

Page 2: It is well known that both approaches provide the same solution at the end of the assimilation period, for perfect and linear models. - -> It is well known that both approaches provide the same solution at the end of the assimilation period, for Gaussian variables, and perfect and linear models.

Modification taken into account

**Page 5, Line 23: index i is forgotten in epsilon**

Modification taken into account

**Page 8, line 32: you said "the second approach was used". I do not understand what is it "the second approach".**

It refers to the type of coding of the modules in the modular graph. Since no detail is given before regarding this pointm the phrase will be erased from the manuscript

**Page 9, line 11: you said "the initial model". Same problem, I do not understand.**

It refers to the reference model, before parameter perturbation

Page 9, line 25: "the parameter prior values were retrieved successfully." In general, we estimate the actual values and not the prior. The prior is what we know initially before observation.

Exactly, but since is a twin experiment our prior is the target value we want to achieve. The phrase will be changed by: **"the assimilation was successfully achieved."**

Page 12, line 24: more difficult it is to find local minima that correspond to the initial control parameters values - -> more difficult it is to find global minima that correspond to the initial control parameters values

Modification taken into account

Page 12, line 25: It is difficult to retrieved parameters - -> It is difficult to retrieve parameters

Modification taken into account

[revised manuscript text omitted]

9 The basic idea is to determine the minimum of a cost function J that measures the misfits between the observations and 10 the model estimations. Due to the complexity of this function, the solution is classically obtained by using gradient 11 methods, which implies the use of the adjoint model of M. This model is derived from the equations of the direct model 12 M. The adjoint model estimates changes in the control variables in response to a disturbance of the output values 13 calculated by M. It is therefore necessary to proceed in the backward direction to the direct model calculations, which 14 means to use the transpose of the Jacobean matrix with respect to the control parameters. When observations are 15 available, the adjoint allows minimizing the cost function J.

Formalism and notations for variational data assimilation are taken from Ide et al., (1997). *M* represents the direct model,  $\mathbf{x}(t_0)$  is the initial state of the model and **k** represents the vector of the inner model parameters to be controlled, so  $\mathbf{x}(t_i) = \mathbf{x}(t_i)$

18  $M_i(\mathbf{k}, \mathbf{x}(t_0))$ , where  $M_i(\mathbf{k}, \mathbf{x}(t_0))$  is represented by  $M \circ M \circ \dots \circ M(\mathbf{k}, \mathbf{x}(t_0))$ . The tangent linear model is noted

19  $\mathbf{M}(t_{i},t_{i+1})$ , which is the Jacobean matrix of  $\mathbf{M}$ , in  $\mathbf{x}(t_{i})$ . The adjoint model  $\mathbf{M}_{i}^{T}$  is the linear tangent transpose, defined as:

20
$$\mathbf{M}_{i}^{T} = \prod_{j=0}^{i-1} \mathbf{M}(t_{j}, t_{j+1})^{T}$$
 Eq. (3)

M is used to estimate variables, which are most often observed from an observation operator **H**, permitting to compare the observed values  $\mathbf{y}^0$  with respect to the **y** calculated by the composition  $\mathbf{H} \cdot \mathbf{M}$ , when they are available. The cost function *J* will be defined in terms of observations, so  $\mathbf{H}_i$  allows us to estimate the variables  $\mathbf{y}_i$ , from the state vector  $\mathbf{x}(\mathbf{t}_i)$ . We suppose that  $\mathbf{y}_i = \mathbf{H}_i(\mathbf{M}_i(\mathbf{x}_i, \mathbf{k})) + \varepsilon_i \mathbf{y}_i = \mathbf{H}_i(\mathbf{M}_i(\mathbf{x}_i, \mathbf{k})) + \varepsilon_i$  where  $\varepsilon_i$  is a random variable with zero mean. This term represents the sum of the model, observation and scaling error. Finally, the most general form of the cost

This term represents the sum of the model, observation and scaling error. Finally, the most general form of the cost function is defined as follows:

27
$$J(\mathbf{k}) = \frac{1}{2} (\mathbf{k} - \mathbf{k}^{b})^{T} \mathbf{B}^{-1} (\mathbf{k} - \mathbf{k}^{b}) + \frac{1}{2} \sum_{i=0}^{t} (\mathbf{y}_{i} - \mathbf{y}_{i}^{0})^{T} \mathbf{R}_{i}^{-1} (\mathbf{y}_{i} - \mathbf{y}_{i}^{0})$$
 Eq. (4)

The background vector is defined as  $\mathbf{k}^{b}$ , which is an *a priori* vector of the inner model parameters. The first part of the cost function represents the discrepancy to  $\mathbf{k}^{b}$  and acts as a regularization term. The second part represents the distance between the observations and the model estimates. **B** is the covariance error matrix of  $\mathbf{k}^{b}$  and  $\mathbf{R}_{i}$  is the covariance error matrix of  $\mathbf{y}^{o}$  at time  $t_{i}$ . The objective of this work is to show the capacity of 4DVAR to help determining the value of the principal inner parameters **k** of SECHIBA and the initial conditions for Surface Water Content. The present distributed software allows the reader to do its own experiments using synthetic or actual data. When the observations are synthetic (produced by the model itself) no transfer function from the estimation to the observation are needed, and **H** is taken as Commentaire [BPHS9]: Modify M. Kallel 1 the identity matrix. If actual data are used, a specific **H** is used that transforms the soil temperature into brightness

2 temperature (see section Model and Data).

The minimization of the cost function (Eq 4) is based on gradient-descent approaches. The cost function gradient has the
 form

5

$$\nabla_{k}J = \mathbf{B}^{-1}(\mathbf{k} - \mathbf{k}^{b}) + \sum_{i=1}^{t} \mathbf{M}_{i}^{T}(\mathbf{k}) \nabla_{yi} f = \mathbf{B}^{-1}(\mathbf{k} - \mathbf{k}^{b}) + \sum_{i=1}^{t} \mathbf{M}_{i}^{T}(\mathbf{k}) \nabla_{yi} J$$
6
Eq (5)

- 7 Where  $\nabla_k J$  and  $\nabla_{vi} J$  are the gradients of the cost function J with respect to **k** and **y**i respectively.
- 8 The expression above allows us to compute  $\nabla_k J$  by knowing  $\nabla_{yi} J$ , in the form of a matrix product of this term by the
- 9 matrix  $\mathbf{M}_{i}^{T}(\mathbf{x},\mathbf{k})$ , corresponding to the transpose of the Jacobian Matrix. The development of calculation gives the
- 10 expression of the gradient of  $y_{:}$  (equation 2):

11
$$\nabla_k J = \mathbf{B}^{-1} \left( \mathbf{k} - \mathbf{k}^b \right) + \sum_{i=1}^t \mathbf{M}_i^T \left( \mathbf{k} \right) H^T \left[ R_i^{-1} \left( y_i - y_0 \right) \right]$$

12 The control parameters are adjusted several times until a stopping criterion is reached. The iterations of the gradient 13 method allow us to approach the solution, in order to satisfy a stopping criterion that could be, for example, a certain 14 threshold on the norm of the cost function gradient.

**15 3.2 YAO**

Variational data assimilation requires the computation of the adjoint code of the direct model, which is a heavy and 16 17 complex task, especially for a large model such as SECHIBA. Usually, the adjoint code is computed with the help of specific softwares (automatic differentiators) (e.g., Bischof et al., 1996; Giering and Kaminski, 2003; Hascoët and 18 19 Pascual, 2004). These softwares are appropriate for the differentiation of large codes, but their use will be optimal only 20 under specific coding conventions and a good level of modularity of the codes (Talagrand, 1991). Moreover, manual 21 optimization of the produced code is often necessary. Therefore, in many practical cases the automatic production of 22 code will not be totally optimal in terms of flexibility (e.g., when the direct model is updated frequently, one has to 23 re-differentiate the whole code). These considerations motivated the development of a slightly different but 24 complementary approach that focuses on the high-level structure of the numerical models, embedding implementation 25 details inside simple entities that can be easily updated. This has led to the development of the YAO assimilation 26 software at LOCEAN/IPSL (https://skyros.locean-ipsl.upmc.fr/~yao/). YAO is based on the decomposition of a 27 numerical model into elementary modules interconnected by directional links. On one side, the structure of the model 28 (variables, dependencies...) is described as a graph structure. On the other side, the details of the physics are coded inside 29 C/C++ basic modules that are ideally simple. The user can therefore separate the "high-level" structure of the model 30 from implementation details. It is also very easy to update a numerical code within this framework. Regarding the 31 assimilation strategy, YAO computes the tangent linear and adjoint codes from the elementary jacobians of each 32 module (provided by the user). Adjoint/cost function test tools are also available. Finally, YAO includes routines devoted to classical assimilation scenario (incremental form ----) and is interfaced with the M1QN3 minimizer (Gilbert 33 34 and Lemaréchal, 1989).

Commentaire [BPHS10]: Modify Mme. Mechri nad M. coudert Code de champ modifié

**Commentaire [BPHS11]:** Modify Mme. Mechri nad M. coudert

Eq (6)

[revised manuscript text omitted]

---

## Editor Decision (ED1)

Many of your replies simply state "Go read my PhD thesis". Nobody will bother reading a whole PhD thesis to find particular information. You need to provide these clarifications. It is not appropriate in a scientific paper to refer to a PhD thesis as much as you do.

Benoit Coudert asked for some further details on the properties of C3 Crops in ORCHIDEE, e.g., LAI, rooting depth, height etc. I think this information should be provided.

I don't understand your reply to the comment about the multiplicative factors. How can you have all parameters (albedo, emissitivity etc) all equal to one? This does not make any sense to me. Were you referring to the multiplicative factor, rather than the actual parameter? If so, what's the point of having a multiplicative factor of one? I don't follow the logic here.

In reply to questions 1 (clarifications of what you mean by first guess and observations) and 2 (about starting dates and decrease in performance) and by Rihab Mechri, you also need to modify the manuscript as other reader may have similar queries. And stop referring to your PhD thesis, provide the information instead.

In response to Abdelaziz Kallel, about the "Gradient Algorithm", "estimation of control parameters", and the third one, clarifications need to be made within the manuscript.

Page 2, paragraph starting with "Variable data assimilation" – This is a rather long paragraph, I suggest breaking it into two.

Page 12, section 4.4, lines 8 to 10 should be one paragraph.

Your results section is very short. The paper does not have a discussion section at all??? You need to relate your work back to the rest of the literature. You have not done this at all in the paper, which I find very odd for a scientific paper.

You state that there is little difference in H and LE because there was no precipitation during the simulation period. You therefore must show results during periods where there is high precipitation. A simulation period of one week is much too short. This must be extended.

You state that your results can be explained by "The complexity of the model" – This is much too broad and general. I expect a discussion of the results to be much more in depth.

It is critical that model evaluation covers a long enough period to sample seasonality, and a range of sites covering a large number of PFTs. You only show results over a one-week period, at one site, and simply refer to your PhD thesis for the Kruger site. This is not appropriate. This paper needs a lot more work.

---

## Author Response (AR2)

**1. REPONSE TO BENOIT COUDERT**

Dear Reviewer,

Thank you for your review and for the interest in our work. I make list of answers regarding all your comments and questions

**SPECIFIC COMMENTS**

**Section 1:**

**P.2, L.31: could you precise what is a "specific deep land surface temperature" ?**

> The sentence has been changed in the revised version because it was wrong and related to another paper not referenced here. The new sentence is now: "When assimilating LST into the model, the authors proved that the assimilation of LST can improve the model simulated heat and water fluxes. "

**P3, L.1: "or" should be replace with "of"**

> Modification taken into account

**P3, L3: remove "available"**

> Modification taken into account

**Section 2:**

**P4., L.34-36: The SECHIBA version used has a "two-layer soil profile" meanwhile in appendix A (P.28, L.8-9) a "seven-layer soil profile" is mentioned for the THERMOSOIL subroutine. Please bring some precisions or corrections.**

> A two-layer hydrology was used in this ORCHIDEE version. The seven layer discretization is for the resolution of the heat diffusion equation. We have changed the text in the paper to make it clearer

**P.4.: L.1-12: could you precise why do you prefer the use of a brightness temperature in the interval [8-14] microns instead of the LST ? I certainly misunderstand the explanation.**

> The use of this variable follows my previous thesis work (Benavides, 2014) when observations coming from a thermal infrared radiometer were used as observations (SMOSREX). This interval correspond to the radiometer filter used for these measurements.

**L.6, Eq.1: the Stefan Boltzmann constant [sigma] has been omitted in the first term of the equation. L.6, Eq.1: is LW_down estimated or measured in situ ? In this case, could you precise the spectral band associated and if a band factor has been applied to take into account that only a fraction of**

**the radiation is measured in the spectral interval according to the Planck's law at the difference of the Stefan-Boltzmann law. Precisions are thus required regarding the use of the Svendsen conversion function ( Eq.2).**

> We don't understand your remark: in equation 1, we wrote the total radiation emitted by a soil surface and integrated on all the long wave spectra. The SB constant don't appear on the left side of the equation. In our case, LW downward is measured by a large band radiometer and this is why we can use the Svendsen's formula to estimate LST. The manuscript has been revised to clarify the notations and the confusions between LST and TB.

**Table 3, P.18: "LST" is mentioned as observation but is it: LST, radiance or brightness temperature in the [8-14] microns interval ? You should also indicate that it is a synthetic observation.**

I can assimilate LST or TB computed from a radiometer measurements. In my distributed version only LST observations are included. In the full SECHIBA-YAO version both measurements can be chosen.

**P.4, L24: could you precise what is the type of the C3 crop for both sites and also give some details on the phenology or state of the plant development. As an example, LAI and canopy height could be added in Table 3 for PFT12.**

Vegetation in ORCHIDEE is characterized by using Plant Functional Type system of classification. Although PFT system describes to types of cultures (C3 and C4crop) it does not distinguish varieties of crops and only one crop type is currently active

**Section 3:**

**P.6, L.3, Eq.5: the cost function "f" should be replace with "J" in relation to Eq.4**

> Modification taken into account

**P.6, L.7: I suppose that "y" should be replaced with "J". I do not understand the reference to equation 2 which is the expression of the brightness temperature**

> Reference to equation 2 misplaced. Modification taken into account

**P.7, L.32: this empty line should be suppressed.**

> Modification taken into account

**P.8, L18-19: the sentence is unclear, please correct the syntax.**

> The phrase will be replaced by: "When studying the subroutines, their complexity was reduced by breaking the different steps into simpler elements."

**P.8, L.32: "the second approach was used" I certainly miss something but you have not presented several approaches in this subsection.**

Misplaced reference: this sentence will be erased

**Section 4: P.9, L.16: "The other parameters are multiplicative factors". Why don't you consider directly the parameters themselves: surface emissivity instead of kemis, albedo instead of kalbedo, etc. ? Is it only due to a technical (or numerical) reason ?**

The idea is to have all parameters with the same value (all equal to 1) , in order to have directly the magnitude of the assimilation quality, and with the idea of having the possibility of comparing them

**P.9, L.23: instead of "optimal value", you certainly mean "initial value" ?**

What I meant is that prior to assimilation and to any perturbation, model parameters are always equal to 1

**P.10, L.5-6 and Table 1 (P.16): the initial value of mxeau (maximum water content) parameter is very low (150kg/m3). Why this choice ?**

This is the initial value generally used in sechiba before spinup.

**What types of soil are considered? It is important to mention somewhere the soil description (classification or texture).**

Yes , you are true, the soil texture has been added in the text .

**A low mxeau value corresponds to dry or stressed surface conditions and will consequently increase the LST and overestimate it compared to in situ measurements. This remark is confirmed by the LE times series of figures 5&6 (see comments below) with quasi null absolute values. Is it done to increase the parameter sensitivity to LST in order to improve the results ?**

Yes , we agree, and this is the case in our experiments , we took dry conditions to be close to the initial value prescribed in Sechiba, but we could have chosen another value. This is at this stage only synthetic observations and twin experiments. The next step is the assimilation of actual observations which will be our future work.

**P.10, L.26-29: in order to facilitate the interpretation of the results of Figure 4 and Table 2, you should precise earlier how the parameter sensitivity hierarchy is defined with both methodologies (finite differences and model gradients), i.e. based on the slope of the gradients.**

I didn't want to give much details on this because I think is out of the scope of the work: However I give a reference to my thesis (Benavides, 2014), where I give much details regarding this remark. However I clarified this point in the final manuscript

**P.11, L.12-18: you should homogenize your notations throughout the text, tables and figures ("true" = observation, "noise" = first guest or perturbed, "assim"= after assimilation) in order to clarify.**

Modification taken into account

**P.12, section 4.4 "Results" and Tables 4 and 5: could you explain how a RMSD on LST reaching 5K is compatible with RMSD on surface fluxes lower than 2.5 W/m2 for experiment 1? The same could be addressed for experiment 2 although RMSD on LST is lower and RMSD on LE higher (but even though relatively low in absolute value). Figures 5&6: times series of LE for bare soil and although for C3 crop have very low absolute values (less than 5W/m2). It is related to the low mxeau value (see previous comment) ? Are the synthetic observations times series realistic compared to real observations ? You should give more information on these points in order to argue your choices and to comment the physical behavior of the model. From a physical point of view, I am surprised by the fact that times series are similar for figures 5 (bare soil) and 6 (C3 crop). During the simulation period of 7 days, LST increases by about 10K meanwhile H flux decrease and LE flux stays quasi null how is it possible ? Times series of meteorological forcing and a description of the vegetation development should be helpful for the analysis.**

The experiments have been done in dry soil conditions , close to the initial value prescribed in Sechiba. We remind that we present here twin experiments, to present the tools developed and their potentialities. The dry soil conditions explain why there is not much difference between bare soil and C3crop with very low evapotranspiration rates. During this period, the ground heat flux increases and heat the soil, explaining the increase of the Surface temperature.

**Section 5: P.13, L.1: "LST" should be replaced with "synthetic LST".**

done

**2. REPONSE TO RIHAB MECHRI**

Dear Reviewer,

Thank you for your review and for the interest in our work. I make list of answers regarding all your comments and questions

**MINOR CORRECTIONS**

• **Abstract**

**1. The sentence corresponding to page 1, lines 16 to 18 is too long and should be shortened or divided in two sentences.**

Remark taken into account. The phrase will be replaced from the manuscript to the following sentence: SECHIBA-YAO allows the control of the eleven most influent internal parameters and the initial conditions of the soil water content. This control is based on the assimilation of land surface temperature observations (in situ or from remote sensing as brightness temperature.

• **Section 2 : Models and Data**

**1. Page 3, line 16 change "22th" to "22nd".**

Modification done

**2. Page 4, line 10 the unit is not clear for the spectral band "um".**

It is the spectral band in micrometers ( μm). Modification taken into account

• **Section 3 : The Methodology**

**–Subsection 3.1: Variational Assimilation**

**1. Page 6, line 3 : you should replace the "f" at the end of equation (5) by "J"**

Modification done

**2. In the page 6, lines 6 and 7 you explained that y is described by equation 2. I can't see the relationship between equation 2 describing the empirical formulation of the brightness temperature and the surface radiation and the description of the observation term "y". Are you making reference to the equation described in page 5 at line 23 ?**

Eq (2) makes reference to the calculation of brithness temperature in SECHIBA based on the empirical formulation of Svendsen et al., 1990. This variable can be later used as observation if remote sensing observations are used. The reference to equation 2 is misplaced, it will be remove from the manuscript

**–Subsection 3.4: Development of SECHIBA-YAO**

**1. Page 8, line 11 : change "ANNEX A" to "Appendix A"**

Modification done

• **Data assimilation experiments**

–**Subsection 4.3: Experiment Definition**

**1. Page 11, line 27 : change as follows : "sensible (H) and latent (LE) heat**

done

–**Subsection 4.4: Experiment Definition**

**1. Page 12, line 25 : correct 'retrieved' to 'retrieve'.**

done

**QUESTIONS AND COMMENTS**

Regarding the questions, I make a point by point answer to all your different comments.

**1. In the variational assimilation can you please specify what do you exactly mean by observations and first guess : what are you exactly assimilating Pnoise ( referred as 'first guess' and 'perturbed' in figures 5 and 6 (a and b)) or Ptrue (referred as 'observations' and 'initial value' in figures 5 and 6 (a and b)) ?**

Since we are performing twin experiments, an initial set of parameters (Ptrue) is used to produce synthetic observations. The idea is to perturbate Ptrue (to obtain Pnoise,meaning my first guess). The idea is to used the synthetic observations produced with Ptrue in order to go from Pnoise to Ptrue by the assimilation process.

**(a) In the case you are assimilation observations then how could you perform your validation using the same observations?**

Since the assimilation process may give control parameters not exactly the same as the parameters wanted, its final values will affect the final model state, thus a comparison between observations and final temperature values can be useful

**(b) In the case you are assimilation your Pnoise then can you explain more how did you perturbed the 'Truth' using your uniform random noise (precise the respective variation ranges of the different assimilated variables so that we can see how much 50% of the nominal value is consistent ) ?**

The perturbation was a random noise produce by the computer, limited up to 50% the true parameter value, equal to one, so the perturbed value is constrained between [0.5 , 1.5]

**2. In the experiment 3 the gal was to show how could the number of variables included in the assimilation affects the performances of the method. In this case Experiment 3 must have the same conditions than Experiment 2 except the number of assimilated variables. Surprisingly you have**

**changed the assimilation period starting the 8th of August 1996 rather than the 3rd of March. My questions are the following :**

**(a) Why did you change the starting date of the assimilation?**

I wanted to test different conditions in the assimilation capabilities. The scope of this work was only to demonstrate the potential of the assimilation tolos developed. For further information and tests, readers can consult my PhD manuscript (Benavides, 2014).

**(b) How could you know that the decrease in the performances is only related to the number of parameters knowing that you have taken a different assimilation period and knowing the fact that the sensitivity of parameters toward LST is - as you have already mentioned- dependent on the seasons, period of the day etc. ?**

The decrease in the performance is related to the complexity of the cost function to minimize: the greater the parameters the more complex will be and a decreased in the assimilation can be expected, regardless the season, period of the day, etc. In my PhD report (Benavides, 2014) I performed other experiments corroborating this statement.

**3. REPONSE TO ABDELAZIZ KALLEL**

Dear Abdelaziz,

Thank you for your review and for the interest in our work. I make list of answers regarding all your comments and questions

**You said in section 3.1 that you use the Gradient algorithm but you do not explain what kind of algorithm it is exactly: is it "Levenberg-Marquardt algorithm" ?**

> Regarding the gradient algorithm, a minimiser called M1QN3 is used within YAO. It use q quasi-Newton technique (the L-BFGS method of J. Nocedal) with a dynamically updated scalar or diagonal preconditioner.

**You do not explain well how to estimate the actual control parameter values given the a priori. Indeed, the relationship prior value/actual value determines the covariance matrix B in Eq. (4)**

> The Eq (4) is the most general form of the variational assimilation. I only give an introduction to this formula, but the estimation for the actual control parameter values are out of the scope of this work.

**In your experiments you do not add noise to observation so in this case R is 0 and Eq.(4) is not well defined (division by 0). For that I suggest to add noise to observation an study the robustness of the developed approach as a function of the noise level.**

> The equation 4 is just the general form. In YAO R is by default the identity matrix so users can modify its value when necessary

**MODIFICATIONS TO THE MANUSCRIPT**

**Page 2: It is well known that both approaches provide the same solution at the end of the assimilation period, for perfect and linear models. - -> It is well known that both approaches provide the same solution at the end of the assimilation period, for Gaussian variables, and perfect and linear models.**

> Modification taken into account

**Page 5, Line 23: index i is forgotten in epsilon**

> Modification taken into account

**Page 8, line 32: you said "the second approach was used". I do not understand what is it "the second approach".**

> It refers to the type of coding of the modules in the modular graph. Since no detail is given before regarding this pointm the phrase will be erased from the manuscript

**Page 9, line 11: you said "the initial model". Same problem, I do not understand.**

> It refers to the reference model, before parameter perturbation

**Page 9, line 25: "the parameter prior values were retrieved successfully." In general, we estimate the actual values and not the prior. The prior is what we know initially before observation**.

> Exactly, but since is a twin experiment our prior is the target value we want to achieve. The phrase will be changed by: **"the assimilation was successfully achieved."**

**Page 12, line 24: more difficult it is to find local minima that correspond to the initial control parameters values - -> more difficult it is to find global minima that correspond to the initial control parameters values**

> Modification taken into account

**Page 12, line 25: It is difficult to retrieved parameters - -> It is difficult to retrieve parameters**

> Modification taken into account

[revised manuscript text omitted]

---

## Author Response (AR3)

Dear Editor,

Thank you for your review and for the interest in our work. I make list of answers regarding all your comments and questions.

Thank you in advance,

Regards,

Hector Benavides
* * *
**Many of your replies simply state "Go read my PhD thesis". Nobody will bother reading a whole PhD thesis to find particular information. You need to provide these clarifications. It is not appropriate in a scientific paper to refer to a PhD thesis as much as you do.**
-Clear explanations of the different references to my PhD thesis are now presented in the manuscript and I no more refer to the thesis

**Benoit Coudert asked for some further details on the properties of C3 Crops in ORCHIDEE, e.g., LAI, rooting depth, height etc. I think this information should be provided.**
-Plant functional types are useful to distinguish the different soil type. In the present case we used the agricultural C3 grass type whose parameters are given in the text. They are:
$V_{cmax, opt}$ (optimal maximum rubisco-limited potential photosynthetic capacity)=90$\mu$mol/m$^{-2}$s$^{-1}$
$T_{opt}$ (Optimum photosyntixc temperature)= 27,5+0,25Tl °C
Tl  (Function of multiannual mean temperature for C3 grasses)
$\square_{max}$(maximum LAI beyond whitch there is no allocation of biomass to leave)=6
$z_{root}$ (exponential depth scale for root length profile)= 0,25m
$\square_{leaf}$ (prescribed leaf albedo)=0,18
h (prescribed height of vegetation) =0,4m
Ac (Critical leaf senescence)=150 days
Ts (weekly temperature beyond which leaves are shed if seasonal temperature trend is negative)=10°C
Hs (weekly moisture stress beyond which leaves are shed)=0,2

**I don't understand your reply to the comment about the multiplicative factors. How can you have all parameters (albedo, emissitity etc) all equal to one? This does not make any sense to me. Were you referring to the multiplicative factor, rather than the actual parameter? If so, what's the point of having a multiplicative factor of one? I don't follow the logic here.**

The parameters are divided into two groups: inner parameters and multiplying factors (Table 1). The first group corresponds to physical parameters. The second group collects parameters weighting some physical processes of SECHIBA. In the initial model, the weighting parameters have the value of one indicating that no weights are used, thus the effect of the assimilation is to allow a local adaptation of these weighting factors.

In order to compare the restitution of each parameter the physical parameters have been normalized. The controlled parameters are scaled by their prior values, so we control nondimentional parameters and a value of one indicates that the variable has been correctly reconstructed. Therefore the reconstructions of all the parameters can be compared.

**In response to Abdelaziz Kallel, about the "Gradient Algorithm", "estimation of control parameters", and the third one, clarifications need to be made within the manuscript.**

A paragraph about the gradient descent has been added in the text section 3.1
*-For the gradient algorithm, a more depth explanation is proposed at the end of section 3.2*
*Finally, YAO includes routines devoted to classical assimilation scenario (incremental form ) and is interfaced with the M1QN3 minimizer (Gilbert and Lemaréchal, 1989). As they metioned, the routine M1QN3 has been designed to minimize functions depending on a very large number of variables, no subject to constraints. The algorithm implements a quasi-Newton technique (L-BFGS) with a dynamically updated scalar or diagonal preconditioner. It uses line-search to enforce global convergence; more precisely, the step-size is determined by the Fletcher-Lemaréchal algorithm and realizes the Wolfe conditions.*

For question 3
The assimilations used no background and a matrix R that is the identity. This was added in the text.
A new experiment has been run that add noise on the observations. For each experiment the conditions of the first guess are mentioned in the text.

in equation 4 , as we have no knowledge about the Matrix R we take the identity matrix indicating that no weight is used during the assimilation process. In the experiments y is just a scalar.

**Page 2, paragraph starting with "Variable data assimilation" – This is a rather long paragraph, I suggest breaking it into two.**
-Consideration taken into account

**Page 12, section 4.4, lines 8 to 10 should be one paragraph.**
-Modification taken into account

**Your results section is very short. The paper does not have a discussion section at all??? You need to relate your work back to the rest of the literature. You have not done this at all in the paper, which I find very odd for a scientific paper.**

-The result section, as well as the different experiments show in the paper were review in order to account for more variability in the experiment parameters : assimilation period, sites, noise added, paraemters, etc.

**You state that there is little difference in H and LE because there was no precipitation during the simulation period. You therefore must show results during periods where there is high precipitation. A simulation period of one week is much too short. This must be extended.**
-A more adapted results to the experiment are known shown

**You state that your results can be explained by "The complexity of the model" – This is much too broad and general. I expect a discussion of the results to be much more in depth.**
-consideration taken into account

**It is critical that model evaluation covers a long enough period to sample seasonality, and a range of sites covering a large number of PFTs. You only show results over a one-week period, at one site, and simply refer to your PhD thesis for the Kruger site. This is not appropriate. This paper needs a lot more work.**
-This aspect is worked on the new version of the manuscript. Other experiments are added and more information regarding the theoretical aspects of my thesis work are mentioned directly on the manuscript

---

## Author Response (AR4)

Dear Editor,

Thank you for your review and for the interest in our work. I make list of answers regarding all your comments and questions.

Thank you in advance,

Regards,

Hector Benavides
* * *
**On Page 2- lines 21 and 25, page 3 line 8, and possibly elsewhere, you use the phrase "In some_reference (yyyy)", which reads a bit odd. Rather, simply rephrase to: "Huang et al. (2003) developed a one-dimensional….", rather than "In Huang et al. (2003) the authors developed a one-dimensional ….". Fix this throughout the manuscript.**

Suggestion taken into account and modified throughout the manuscript

**Page 4, line 9, should be "Dufresne et al. (2013)".**

Modification taken into account

**Page 4, line 19, years need to be in brackets for in text references???**

Modification taken into account

**Section 4.4.3 – Merge these short paragraphs into one or two paragraphs.**

Modification taken into account

**What is the point of showing Multiplying factors in Table 1, when these all have a value of 1. Can't you just state that in the text? What's the point of showing that you use the same multiplying factor for all these parameters in a Table? This should be removed from Table 1 as it's not providing any useful information. You've already stated this in the text.**

Modification taken into account. In the new version of the manuscript, Table 1 mention only parameters, multiplying factor are stated in the text.

**Figure 3 – What is this "REMPLACER CHAPEAU PAR HAT"????**

Transcription error, the phrase is removed from the manuscript